# Light-induced Floquet spin-triplet Cooper pairs in unconventional magnets

Pei-Hao Fu[1*], Sayan Mondal[2], Jun-Feng Liu[1] and Jorge Cayao[2†]

**1** School of Physics and Materials Science, Guangzhou University, Guangzhou 510006, China
**2** Department of Physics and Astronomy, Uppsala University, Box 516, S-751 20 Uppsala, Sweden

★ phy.phfu@gmail.com , † jorge.cayao@physics.uu.se

## Abstract

The recently predicted unconventional magnets offer a new ground for exploring the formation of nontrivial spin states due to their inherent nonrelativistic momentum-dependent spin splitting. In this work, we consider unconventional magnets with $d$- and $p$-wave parities, and investigate the effect of time-periodic light drives for inducing the formation of spin-triplet phases in the normal and superconducting states. In particular, we consider unconventional magnets without and with conventional superconductivity under linearly and circularly polarized light drives and treat the time-dependent problem within Floquet formalism, which naturally unveils photon processes and Floquet bands determining the emergent phenomena. We demonstrate that the interplay between unconventional magnetism and light gives rise to a non-trivial light-matter coupling which governs the emergence of Floquet spin-triplet states with and without superconductivity that are absent otherwise. We find that photon-assisted processes promote the formation of spin densities and spin-triplet Cooper pairs between different Floquet sidebands. More precisely, the Floquet sidebands offer an additional quantum number, the Floquet index, which considerably broadens the classification of superconducting correlations that lead to Floquet spin-triplet Cooper pairs as an entirely dynamical phenomenon due to the interplay between light and unconventional magnetism. Furthermore, we discuss how the number of photons is connected to the symmetry of Cooper pairs and also explore how the distinct light drives can be used to manipulate them and probe the angular symmetry of unconventional magnets. Our results therefore unveil the potential of unconventional magnets for realizing nontrivial light-induced superconducting states.

# 1   Introduction

Unconventional magnets have recently emerged as a new class of magnetic systems beyond the conventional dichotomy of ferromagnets and antiferromagnets [1–6]. This so-called third class of magnetism exhibits spin-split Fermi surfaces similar to ferromagnets [7,8], yet maintains zero net magnetization due to compensated magnetic ordering akin to antiferromagnets [9–11]. Interestingly, depending on the momentum-space parity of their magnetization, unconventional magnets can be classified into various angular momentum channels that include even- and odd-parity symmetries [4,6]; see also Refs. [12–14] for prior works. Among the odd-parity unconventional magnets, we find e.g., $p$- and $f$-wave, while $d$-, $g$-, and $i$-wave for even-parity magnets; these even-parity magnets are often called altermagnets [15,16], while odd-parity magnets are simply referred to as either $p$-wave [17,18] or $f$-wave magnets [17] depending on their symmetry; see also Refs. [19–26] for recent studies. Furthermore, unconventional magnets with even-parity (e.g., $d$-, $g$-, and $i$-wave) break both rotational and time-reversal symmetry but preserve their combined symmetry, while odd-parity magnets (e.g., $p$- and $f$-wave) only break rotational symmetry with the time-reversal symmetry preserved [4,6]. These unique properties of unconventional magnets have motivated several studies since their prediction [1–6], with exotic phenomena that include the prediction of a giant magnetoresistance [27], anomalous Hall effects [28–33], spin-orbit torques [34,35], spin filtering effects [4,36–47], strongly correlations in Mott insulators [48], non-linear transport [22], non-Hermitian effects [49,50], multipoles [51], spin Edelstein [52] among other

phenomena [53–57]; see Refs. [1–6] for recent reviews. With numerous candidate materials, including $RuO_2$ [28, 31, 58, 59], MnTe [30, 60], $Mn_5Si_3$ [32, 61], CrSb [62], and $Mn_2Au$ [63], exploring unconventional magnets open a new avenue for realizing magnetic phenomena in systems without net magnetization.

The promising properties of unconventional magnets have made a particular impact when combining with superconductors [6, 64], where a plethora of unconventional superconducting phases have been reported [65–81]. One of the key consequences of the interplay between superconductivity and unconventional magnetism is the appearance of superconducting correlations defining Cooper pairs with spin and angular properties inherent to unconventional magnets [6, 65, 72–75, 77, 82–85]. In this regard, unconventional magnets with superconductivity not only induce a spin-singlet to spin-triplet conversion but also transfer their parity to the emergent superconducting correlations, giving rise to Cooper pairs with spin-triplet symmetry and higher angular momentum [6, 72, 74, 75]. In junctions with conventional $s$-wave superconductors, unconventional magnets have been shown to produce novel effects such as the superconducting diode effect [86–91], nonlinear superconducting magnetoelectric effects [92, 93], superconducting spin-splitter behavior [94], and various Josephson effects [74, 95–102] related to Andreev reflections [103–110]. Furthermore, unconventional magnets have been used for realizing Majorana zero modes in topological superconductors [111–114] as well as in higher-order topological superconductors [115, 116] without the need for external magnetic fields [71, 111, 117–121]. All these emergent phases largely stem from the nontrivial spin-momentum coupling in unconventional magnets, suggesting that coupling them to external fields could give rise to even more exotic states.

One prominent example of external fields that uniquely couple to matter is light, which not only enables control over intrinsic properties but also drives the emergence of novel non-equilibrium states that are absent in the static regime. Of particular interest are time-periodic light drives, which, through the framework of Floquet theory [122–124], provide a powerful approach to design dynamic phases via Floquet engineering [125–127]. In the normal state, the application of light-drives via Floquet engineering has permitted the realization of light-induced Hall effects [128–135], Floquet topological insulators [136–145], light-induced Weyl semimetals [146–148], Floquet time crystals [149–152], among other dynamic phases [125–127, 153–156]. Moreover, Floquet engineering in the normal state has also enabled ultrafast control of spin and magnetic textures on picoseconds or sub-picoseconds timescales, giving rise to the field of ultrafast spintronics [157–161]. In superconducting systems, light fields within Floquet theory have also been proven to be a key tool to design unique non-equilibrium phenomena. For instance, this involves Floquet Majorana modes [162–174], Floquet Majorana time-crystals [162–164, 175], the detection of topological phase transitions via the Josephson effect [176, 177], Floquet-Andreev states [178–183], light-controlled Higgs modes [184, 185], dynamic spin-triplet superconductivity [186], and Floquet Cooper pairs [184, 187]. Very recently, the effect of light drives on unconventional magnets with superconductivity has also been addressed, but then the few existent studies mostly consider high-frequency light drives [75, 188]. As a result and despite all these advances, the role of Floquet sidebands on emergent light-induced superconducting effects in unconventional magnets remains largely unexplored.

In this work, we consider time-periodic light drives in unconventional magnets with spin-singlet $s$-wave superconductivity and investigate the emergence of light-induced Floquet Cooper pairs. In particular, we consider $p$- and $d$-wave unconventional magnets under time-periodic circularly and linearly polarized light drives, as schematically shown in Fig. 1. We first discuss how unconventional magnetism induces a spin density and spin-triplet Cooper pairs in the static regime of normal and superconducting states (Fig. 2). We then demonstrate that the interplay between light and unconventional magnetism gives rise to a nontrivial light-matter

interaction that is the key for realizing light-induced Floquet spin-triplet states, which are absent in the static regime. We unveil that these Floquet spin-triplet states appear as a result of the Floquet sidebands, hence involving single- and two-photon assisted processes. In the normal state, the Floquet spin density can be directly used to identify the strength of unconventional magnetism (Fig. 3 and Fig. 4). We find that this is also possible in the superconducting state and, additionally, in this case, we obtain that the Floquet spin-density is directly tied to the emergence of Floquet Cooper pairs with spin-singlet and spin-triplet configurations. These Floquet Cooper pairs are allowed due to the extra quantum number, the Floquet sideband index, which broadens the classification of superconducting symmetries into Floquet classes that coexist between different Floquet sidebands through the absorption or emission of photons; see Fig. 5 and Table 1. Among the Cooper pairs symmetry classes, we identify two types of Floquet spin-singlet Cooper pairs induced purely due to the effect of the light field on the parent spin-singlet superconductor, and two emergent types of Floquet spin-triplet Cooper pairs arising entirely due to the combined interaction of light, unconventional magnetism, and spin-singlet superconductivity.

We further show that the number of photons involved in the formation of Floquet Cooper pairs is crucial for classifying the different pairing classes (Fig. 6), which reveals distinct parity pairs in driven $d$-wave (Fig. 7) and $p$-wave magnets (Fig. 8). This also permits us to conclude that all pair amplitudes involving an odd number of photons are entirely induced by the light drive, while those involving an even number of photons can exist even in the static phase due to their link to the parent superconductor (Figs. 9 and 10). Finally, we obtain that the Cooper pair amplitude can be controlled through the competition between the polarization direction of linearly polarized light and the orientation of the unconventional magnetic order, providing a potential approach for identifying the angular symmetry of unconventional magnetism in both the high-frequency [75] and the low-frequency regimes (Fig. 11). We stress that, while our results focus on $p$-wave and $d$-wave unconventional magnets, the presented methodology and outcome are applicable to other types of unconventional magnets, such as those with higher angular momentum symmetries (Appendix A) and higher unconventional magnetic fields (Appendix B). Our findings demonstrate that unconventional magnets are rich systems for realizing light-induced superconducting states by means of Floquet engineering.

This paper is organized as follows. In Sec. 2, we present the models for unconventional magnets with and without spin-singlet $s$-wave superconductivity in the static regime; here we examine the spin density and spin-triplet Cooper pairs, with a particular focus on $d$-wave and $p$-wave unconventional magnets. In Sec. 3, we analyze the nontrivial light–matter interactions and their Floquet description in unconventional magnets with and without superconductivity. Sec. 4 and 5 present a detailed analysis of the Floquet spin density and Floquet Cooper pairs. The conclusions are summarized in Sec. 7. In appendices A and B, we discuss higher angular momentum symmetries and $d$-wave altermagnets at stronger exchange fields, respectively.

## 2 Unconventional magnets in the static regime without and with superconductivity

In this section, we introduce the simplest models to describe unconventional magnets with and without spin-singlet $s$-wave superconductivity in the static regime. In both cases, we discuss how the unconventional magnetism induces a spin density and, in the superconducting part, we additionally show the emergence of spin-triplet Cooper pairs; these results are summarized in Fig. 2.

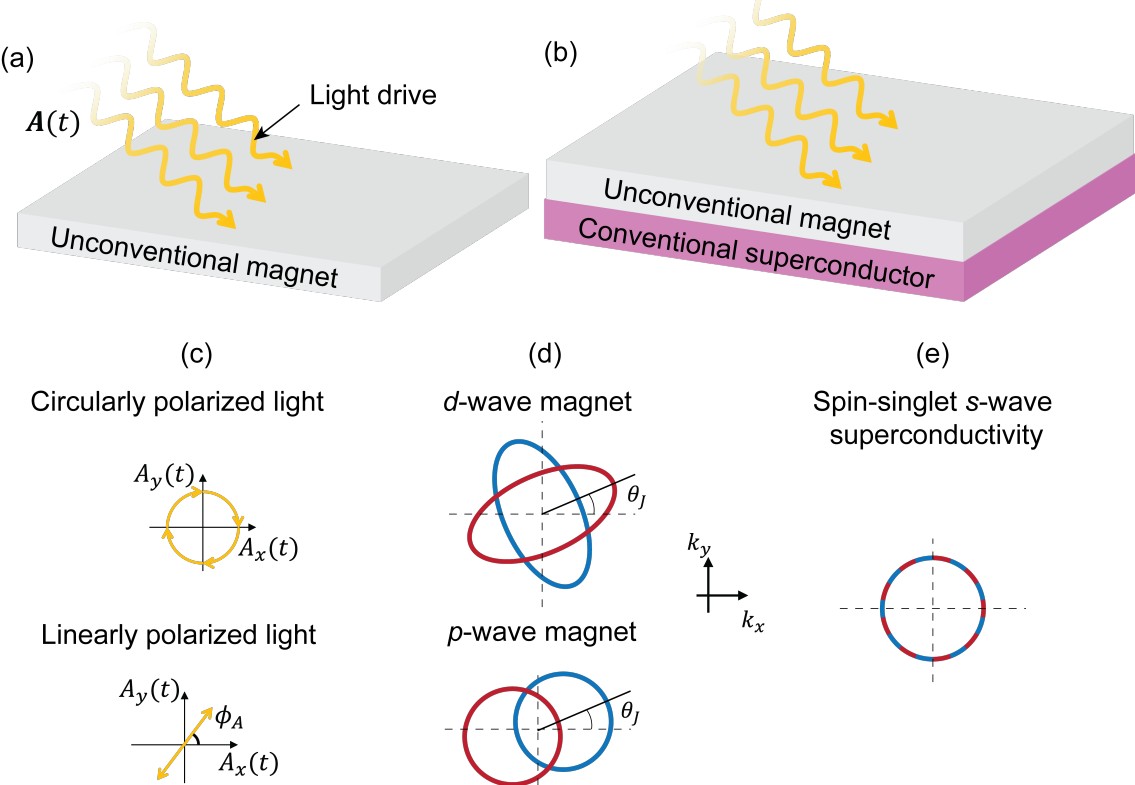

Figure 1: (a) Schematics of an unconventional magnet (gray) under a time-periodic light drive $A(t)$ (wiggle yellow arrows). (b) An unconventional magnet in proximity to a conventional spin-singlet $s$-wave superconductor (violet) under a time-periodic light drive. (c) Illustration of circularly and linearly polarized light drives, with components $A_{x,y}$. Here, $\phi_A$ denotes the linear polarization angle. (d) Spin-split Fermi surfaces in the $k_x$–$k_y$ plane for $d$-wave and $p$-wave unconventional magnets, with $\theta_J$ representing the orientation angle measured from the $x$-axis. The blue and red colors correspond to up and down spins. (e) Sketch of the isotropic order parameter of a conventional spin-singlet $s$-wave superconductor.

## 2.1 Unconventional magnets in the normal state

To model unconventional magnets, we consider the following low-energy Hamiltonian [6]

$$H_q^\sigma(\boldsymbol{k}) = \xi_{\boldsymbol{k}} + \sigma J_q(\boldsymbol{k}), \tag{1}$$

where the Néel vector is chosen along $z$, $\sigma = +1$ ($-1$) denotes the spin-up (spin-down), $\boldsymbol{k} = (k_x, k_y)$, $\xi_{\boldsymbol{k}} = Bk^2 - \mu$ is the kinetic energy with $k = \sqrt{k_x^2 + k_y^2}$ and the chemical potential $\mu$. Moreover, $J_q$ is the field characterizing unconventional magnetism and described by [6]

$$J_q(\boldsymbol{k}) = \alpha_q k^q \cos\left[q\left(\theta_k - \theta_J\right)\right], \tag{2}$$

where $\alpha_q$ represents the strength of unconventional magnetism, $\theta_k = \arctan\left(k_y/k_x\right)$, and $\theta_J$ is the magnetic direction, see Fig. 1(d). Eq. (2) is anisotropic depending on the crystalline momentum orientation $\theta_k$ and $\theta_J$. Different unconventional magnets are categorized by the order of momentum, $q$, in Eq. (2). In particular, for $q = \{0, 1, 2, 3, 4, 6\}$, the field $J_q(\boldsymbol{k})$ describes $s$-wave, $p$-wave, $d$-wave, $f$-wave, $g$-wave, or $i$-wave magnets, respectively; see Refs. [5, 6, 15–19, 66] and App. A. The $s$-wave case, with $q = 0$, is a conventional ferromagnet breaking time-reversal symmetry $\mathcal{T}$. In other $q$-even magnets, time-reversal symmetry

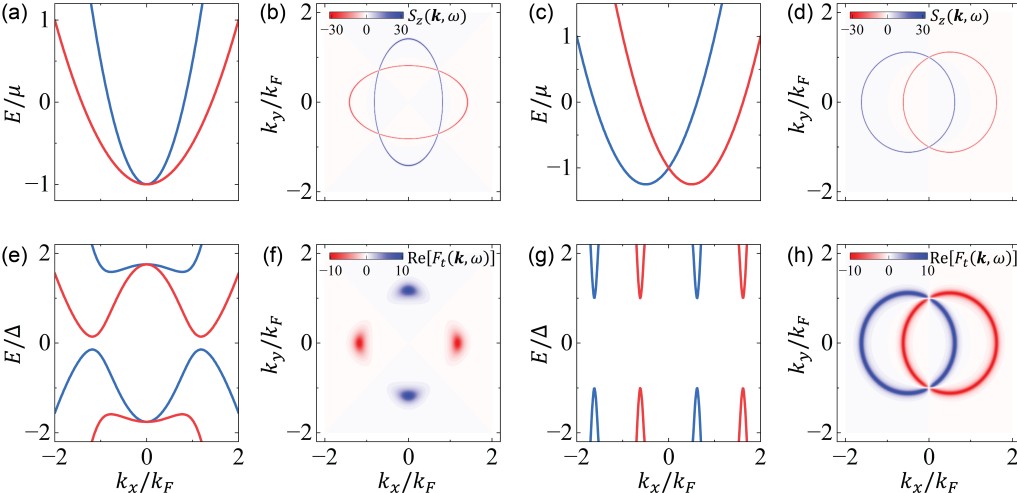

Figure 2: (a,b) Energy dispersion (a) and spin density (b) for a $d_{x^2-y^2}$-wave alter-magnet in the normal state ($\Delta = 0$) at $k_y = 0$, while in (c,d) the same is plotted but for a $p_x$-wave magnet. The blue and red colors in (a,c) indicate spin-up and spin-down bands, respectively, while in (b,d) the blue and red colors indicate the positive and negative values of the spin density. In (b,d), the frequency is chosen as $z = 0 + i10^{-3}$. (e,f) Energy dispersion and real part of the spin-triplet pair amplitude for a $d_{x^2-y^2}$-wave altermagnet with spin-singlet $s$-wave superconductivity. (g,h) The same quantities as in (e,f) but for a $p_x$-wave magnet with spin-singlet $s$-wave super-conductivity. In (e,g), the superconducting energy bands formed by spin-up electrons and spin-down holes are depicted in blue, while the bands formed by spin-down elec-trons and spin-up holes are shown in red. The blue and red colors in (f,h) mark the positive and negative values of the real part of the spin-triplet pair amplitude; here $z = 0.1\Delta + i10^{-3}$. Parameters: $B = 1$, $\alpha_d = \alpha_p = 0.5$, $\theta_J = 0$, $\mu = 1$; in (e-h) we also consider $\Delta = 0.7\mu$ for the $d_{x^2-y^2}$-wave altermagnet and $\Delta = 0.1\mu$ for $p_x$-wave magnet.

$\mathcal{T}$ and rotational symmetry $\mathcal{C}_{2q}$ are broken, but the joint operation of $\mathcal{T}$ and $\mathcal{C}_{2q}$ preserved characterizing the so-called altermagnetism [15, 16, 66]. While the $q$-odd magnets in their simplest forms only break $\mathcal{C}_{2q}$ but preserve $\mathcal{T}$ due to the coupling between spin and odd-order momentum, which are considered as $p$- and $f$-wave unconventional magnetism [17, 18]. Con-sidering more complicated models with noncoplanar spins and distinct sublattices, however, time-reversal symmetry is broken [6, 17, 18]. Nevertheless, Eq. (2) is sufficient for describ-ing the parity of the magnetization and the resulting anisotropic spin splitting [6, 18]. It is worth noting that Eq. (2) has a nonrelativistic origin [4, 6, 15, 17, 18, 66], even though it is akin to the common relativistic spin-orbit coupling [189, 190]; this implies that the spin-splitting in unconventional magnets can be much larger than those due to the relativistic spin-orbit coupling [4].

As noted above, a key unconventional magnetic feature captured by Eq. (2) is the mo-mentum dependence of the field, which leads to an anisotropic spin splitting in the bands, see below. Under general conditions, for $\theta_k = \theta_q + n\pi/q$ ($n \in \mathbb{Z}$), the unconventional magnetic effect is most pronounced and the spin-splitting depends solely on $\alpha_q k^q$, while it vanishes for $\theta_k = \theta_q + (2n + 1)\pi/(2q)$ and gives spin-degenerate bands. To further inspect the spin split-ting in the bands, we focus on two typical unconventional magnets having $p$-wave ($q = 1$) and $d$-wave ($q = 2$) parity, which are often referred to as $p$-wave magnets [17, 18] and $d$-wave altermagnets [15], respectively. For completeness, in App. A, we also explore unconventional

magnets with higher angular momentum. Hereafter, we focus on $d$- and $p$-wave unconventional magnets. Thus, by expanding Eq. (2), Eq. (1) for $p$- and $d$-wave unconventional magnets are given by

$$H_{p,d}^{\sigma}(\boldsymbol{k}) = \xi_{\boldsymbol{k}} + \sigma J_{p,d}(\boldsymbol{k}), \tag{3}$$

where $J_{d,p}(\boldsymbol{k}) \equiv J_{1,2}(\boldsymbol{k})$ are obtained from Eq. (2) and read

$$
\begin{aligned}
J_d(\boldsymbol{k}) &= \alpha_d \left[ 2k_x k_y \sin 2\theta_J + \left( k_x^2 - k_y^2 \right) \cos 2\theta_J \right], \\
J_p(\boldsymbol{k}) &= \alpha_p \left[ k_x \cos \theta_J + k_y \sin \theta_J \right].
\end{aligned}
\tag{4}
$$

Here, we have two types of $d$-wave altermagnets: for $\theta_J = \pi/4$, we have a $d_{xy}$-wave altermagnet; for $\theta_J = 0$, we have a $d_{x^2-y^2}$-wave altermagnet. Similarly, for $p$-wave magnets at $\theta_J = 0$ we have a $p_x$-wave magnet, while a $p_y$-wave magnet for $\theta_J = \pi/2$.

To further assess the spin splitting in unconventional magnets modeled by Eq. (3) and Eq. (4), we obtain the eigenvalues of Eq. (3), which are given by

$$E_{d,p}^{\sigma}(\boldsymbol{k}) = \xi_{\boldsymbol{k}} + \sigma J_{d,p}(\boldsymbol{k}) \tag{5}$$

These energies are plotted in Figs. 2(a,c) for a $d_{x^2-y^2}$-wave magnet and a $p_x$-wave magnet, respectively, which demonstrate the momentum-dependent spin splitting of energy bands. As already discussed before, the bands are degenerate in spin when $\theta_k = \theta_J + (2n+1)\pi/4$ for $d$-wave altermagnets and $\theta_k = \theta_J + (2n+1)\pi/2$ for $p$-wave magnets, hence further unveiling the anisotropic spin splitting. Moreover, the momentum-dependent spin splitting and spin texture can be revealed in the spin density along $z$, which can be obtained as

$$S_z(\omega, \boldsymbol{k}) \equiv -\frac{1}{\pi} \mathrm{Im} \mathrm{Tr}[\sigma_z G(\omega + i0^+, \boldsymbol{k})] = \frac{1}{\pi} \mathrm{Im} \frac{2J_{d,p}(\boldsymbol{k})}{J_{d,p}^2(\boldsymbol{k}) - (\omega + i0^+ - \xi_{\boldsymbol{k}})^2}, \tag{6}$$

where $G(z, \boldsymbol{k}) = [z - H_{d,p}(\boldsymbol{k})]^{-1}$, with $H_j = \mathrm{diag}(H_j^+, H_j^-)$, $j = (d, p)$, and $z = \omega + i0^+$ defines the retarded Green's function. Eq. (6) clearly shows that the spin-triplet spin density $S_z(\omega, \boldsymbol{k})$ is directly determined by unconventional magnetism via $J_{d,p}(\boldsymbol{k})$. To see the effect of $J_{d,p}(\boldsymbol{k})$, in Figs. 2(b,d) we show $S_z(\omega, \boldsymbol{k})$ as a function of momenta for $d_{x^2-y^2}$- and $p_x$-wave unconventional magnets. We clearly observe that the spin densities develop $d$- and $p$-wave symmetries, with spin-polarized Fermi surfaces, thus revealing a key feature of unconventional magnets. We stress that, since Eq. (6) is not limited to $d$- and $p$-wave unconventional magnets, the behavior of the spin density is expected to characterize the parity in other angular momentum unconventional magnets, such as $f$-, $g$- and $i$-wave magnet, which we further discuss in App. A.

## 2.2 Unconventional magnets with conventional superconductivity

Having discussed unconventional magnets in the normal state, this part addresses them when conventional spin-singlet $s$-wave superconductivity is induced by the proximity effect, see Fig. 1(b,d,e). We model this system by the following Bogoliubov-de Gennes (BdG) Hamiltonian

$$
\mathcal{H}_q^{\nu}(\boldsymbol{k}) = \xi_{\boldsymbol{k}} \tau_z + \nu \Delta \tau_x +
\begin{cases}
\nu J_q(\boldsymbol{k}) \tau_0, & q\text{-even}, \\
\\
\nu J_q(\boldsymbol{k}) \tau_z, & q\text{-odd},
\end{cases}
\tag{7}
$$

where $\nu = +(-)$ for the Nambu spinor $(\psi_{\boldsymbol{k},\uparrow(\downarrow)}, \psi_{-\boldsymbol{k},\downarrow(\uparrow)}^{\dagger})^T$, while $\tau_i$ is the $i$-th Pauli matrix in Nambu space. The BdG spectrum is then given by

$$
E_q^{\nu,\gamma}(\boldsymbol{k}) =
\begin{cases}
\nu J_q(\boldsymbol{k}) + \gamma \sqrt{\xi_{\boldsymbol{k}}^2 + \Delta^2}, & q\text{-even}, \\
\\
\gamma \sqrt{\left[ \xi_{\boldsymbol{k}} + \nu J_q(\boldsymbol{k}) \right]^2 + \Delta^2}, & q\text{-odd},
\end{cases}
\tag{8}
$$

where $\gamma = \pm 1$.

In Fig. 2(e,g), we plot $E_q^{\nu,\gamma}(\boldsymbol{k})$ as a function of momentum for a $d_{x^2-y^2}$- and $p_x$-wave magnet with conventional superconductivity. Already from the energy dispersions, one can draw some interesting conclusions about the nature of Cooper pairs in unconventional magnets. In the $q$-even unconventional magnets with superconductivity, spin-dependent finite momentum Cooper pairs [191, 192] are formed due to different momenta between the paired electrons and holes with opposite spin. This can be seen in the BdG spectrum of a $d_{x^2-y^2}$-wave magnet with superconductivity in Fig. 2(e), where spin-dependent finite Cooper pair momentum is given by $K_C^\nu = k_e^\nu - k_h^{-\nu}$, with $k_\tau^\nu = \sqrt{\mu/(B + \tau \nu \alpha_d)}$ determined by the zero-energy momentum of electrons and holes with $\tau = +1$ and $\tau = -1$, respectively. Such spin-dependent finite momentum moves the superconducting gap centers resulting in Doppler energy shifts [191, 192], which can lead to a gapless superconductor with hybridization gaps [68, 193, 194] when the superconducting order parameter is below the critical value $\Delta_c = \alpha_d k_e^\nu k_h^{-\nu} = \alpha_d \mu / \sqrt{B^2 - \alpha_d^2}$. On the contrary, in the $q$-odd case, there is no shift in the superconducting gap center due to the absence of Doppler energy shifts as demonstrated in Fig. 2(g). In this case, although the spin-degeneracy is lifted, time-reversal symmetry is preserved due to the odd-order momentum term, resulting in zero-momentum Cooper pairs. Consequently, the superconducting gap remains centered at zero energy, similar to non-magnetic superconducting systems.

A key consequence of the superconducting state is the formation of Cooper pairs, which are characterized by superconducting pair amplitudes [195–201]. These pair amplitudes can be directly obtained from the anomalous electron-hole Green's function, which corresponds to the off-diagonal elements of the Green's function associated with the BdG Hamiltonian

$$\mathcal{G}^\nu(z, \boldsymbol{k}) = [z - \mathcal{H}_q^\nu(\boldsymbol{k})]^{-1} = \begin{pmatrix} G^\nu(z, \boldsymbol{k}) & F^\nu(z, \boldsymbol{k}) \\ \bar{F}^\nu(z, \boldsymbol{k}) & \bar{G}^\nu(z, \boldsymbol{k}) \end{pmatrix}. \tag{9}$$

with $z$ representing complex frequencies and $\mathcal{H}_q^\nu(\boldsymbol{k})$ given by Eq. (7). Moreover, $G^\nu(z, \boldsymbol{k})$ and $\bar{G}^\nu(z, \boldsymbol{k})$ are the normal Green's functions, while $F^\nu(z, \boldsymbol{k})$ and $\bar{F}^\nu(z, \boldsymbol{k})$ are the anomalous Green's functions. While the normal Green's functions enable to calculation of the density of states and spin densities, the anomalous Green's functions characterize the superconducting pair amplitudes [120, 198–201]. In this regard, the functional dependence of the pair amplitudes on the present quantum numbers determines the symmetry of emergent superconducting correlation [120, 198–201], which must be antisymmetric under the total exchange of quantum numbers as dictated by Fermi-Dirac statistics. To identify the spin symmetry of the Cooper pairs we calculate the even and odd combination in spin indices as $F^{s(t)}(z, \boldsymbol{k}) = [F^+(z, \boldsymbol{k}) \mp F^-(z, \boldsymbol{k})]/2$ since $F^\nu(z, \boldsymbol{k})$ with $\nu = + (-)$ represents the pairing between spin up (down) and down (up) fermions. Thus, we find spin-singlet and spin-triplet pair amplitudes given by [6, 72]

$$F^s(z, \boldsymbol{k}) = \frac{\Delta}{D(z)} \left[ z^2 - \xi_{\boldsymbol{k}}^2 - \Delta^2 + (-1)^q J_q^2(\boldsymbol{k}) \right], \tag{10}$$

and

$$F^t(z, \boldsymbol{k}) = \frac{2\Delta J_q(\boldsymbol{k})}{D(z)} \times \begin{cases} z, & q\text{-even}, \\ -\xi_{\boldsymbol{k}}, & q\text{-odd}, \end{cases}, \tag{11}$$

where

$$D(z) = \prod_{\sigma, \gamma} \left[ z - E_q^{\sigma, \gamma}(\boldsymbol{k}) \right]. \tag{12}$$

is an even function in $z$. Thus, spin-singlet and spin-triplet superconducting correlations co-exist in unconventional magnets with spin-singlet $s$-wave superconductivity. While the spin-singlet pairing results from the parent superconductor being spin-singlet $s$-wave, the spin-triplet component arises entirely due to the interplay between unconventional magnetism and superconductivity. In fact, the spin-triplet pairing is directly proportional to the field $J_q(\boldsymbol{k})$ of the unconventional magnet [Eq. (11)], which, besides influencing the spin-singlet to spin-triplet conversion, also plays a key role on the parity and frequency symmetries. This can be seen by noting that the spin-triplet pair amplitude $F^t(z, \boldsymbol{k})$ in Eq. (11) can be even or odd in momentum $\boldsymbol{k}$ (as well as in frequency $z$) depending on $J_q(\boldsymbol{k})$. On one hand, for $q$-even, $J_q(\boldsymbol{k}) = J_q(-\boldsymbol{k})$, which leads to a spin-triplet pair amplitude with even-parity and odd-frequency. On the other hand, for $q$-odd, $J_q(\boldsymbol{k}) = -J_q(-\boldsymbol{k})$, implying that the spin-triplet pair amplitude is odd-parity and even-frequency. The spin-triplet Cooper pairs in $p$- and $f$-wave magnets with spin-singlet $s$-wave superconductivity exist with time-reversal symmetry, which is akin to the cases of Rashba superconductors [6, 201–205]. For a $d$-wave altermagnet ($q = 2$), the induced spin-triplet pair amplitude has odd-frequency spin-triplet $d$-wave symmetries, while even-frequency spin-triplet even-parity for a $p$-wave magnet ($q = 1$). Hence, the momentum parity of the unconventional magnet is transferred to the emergent spin-triplet pair amplitude [6, 72]. It is worth noting that, while odd-frequency spin-triplet Cooper pairs have been studied before [172, 204, 206–221], such studies did not addressed the momentum parity transfer as it happens in unconventional magnets [6, 72]. To further visualize the momentum dependence of the spin-triplet pair amplitude $F^t(z, \boldsymbol{k})$, in Figs. 2(f,h) we plot its real part a function of $k_x$ and $k_y$ for a $d_{x^2-y^2}$- wave altermagnet and $p_x$-wave magnet, respectively. The first feature to notice is that $F^t(z, \boldsymbol{k})$ exhibits the parity of the underlying unconventional magnet, see also the normal state spin-densities in Figs. 2(b,d). We can therefore conclude that, although the parent superconductor is spin-singlet $s$-wave, unconventional magnetism is able to induce spin-triplet Cooper pairs as well as spin densities as two important characteristic effects. In the next sections, we explore how these two properties respond to the effect of a time-periodic light drive.

# 3  Nontrivial light-matter interaction in unconventional magnets and its Floquet description

In this section, we explore the effect of applying a time-periodic light drive on unconventional magnets without and with spin-singlet $s$-wave superconductivity. We consider two types of time-periodic light drives, involving circularly polarized light (CPL) and linearly polarized light (LPL), in systems with $d$-wave and $p$-wave magnetic order. Moreover, we show how these time-periodic systems are described within Floquet formalism in an extended frequency space.

## 3.1  Driven unconventional magnets in the normal state

We consider time-periodic CPL and LPL drives applied to unconventional magnets. The effects of CPL and LPL are described, respectively, by time-dependent vector potentials as

$$\boldsymbol{A}_C(t) = \frac{E_0}{\Omega} (\cos \Omega t, \eta \sin \Omega t), \tag{13}$$

$$\boldsymbol{A}_L(t) = \frac{E_0}{\Omega} \cos \Omega t (\cos \phi_A, \sin \phi_A), \tag{14}$$

where $\eta = +1$ ($-1$) indicates right-handed (left-handed) circular polarization [222], $\phi_A$ represents the real-space polarization direction [223] as denoted in Fig. 1(c), $E_0$ is the field

amplitude, and $\Omega = 2\pi/T$ the frequency of the drive with period $T$. Then, the effect of the light drives is incorporated in the Hamiltonian describing the unconventional magnets [Eq. (3)] via the minimal coupling or Pierls substitution [224] as $\boldsymbol{k} \to \boldsymbol{k} + e/\hbar \boldsymbol{A}(t)$, where $e$ is the elementary charge. This leads to a time-dependent Hamiltonian $H_q^\sigma(\boldsymbol{k}, t) = H_q^\sigma[\boldsymbol{k} + e/\hbar \boldsymbol{A}(t)]$, which, for $d$-wave magnet and $p$-wave unconventional magnets, reads

$$H_{j,\beta}^\sigma(\boldsymbol{k}, t) = H_j^\sigma(\boldsymbol{k}) + V_{j,\beta}^\sigma(t), \tag{15}$$

where $j = \{p, d\}$ denotes the type of unconventional magnet, $H_j^\sigma(\boldsymbol{k})$ describes the static unconventional magnets and is given by Eq. (3), while $V_{j,\beta}^\sigma(t)$ characterizes the effects of the drives with $\beta = \{\text{CPL}, \text{LPL}\}$ in $d$- and $p$-wave unconventional magnets. For $d$-wave altermagnets under CPL and LPL, $V_{d,\beta}^\sigma(t)$ is given by

$$
\begin{aligned}
V_{d,\text{CPL}}^\sigma(t) &= 2Bk_A k \cos(\theta_k - \eta\Omega t) \\
&\quad + 2\sigma\alpha_d k_A k \cos(\theta_k - 2\theta_J + \eta\Omega t) \\
&\quad + \sigma\alpha_d k_A^2 \cos(2\theta_J - 2\eta\Omega t),
\end{aligned}
\tag{16}
$$

$$
\begin{aligned}
V_{d,\text{LPL}}^\sigma(t) &= 2Bk_A k \cos(\theta_k - \phi_A)\cos\Omega t + Bk_A^2 \cos^2\Omega t \\
&\quad + 2\sigma\alpha_d k_A k \cos(\theta_k + \phi_A - 2\theta_J)\cos\Omega t \\
&\quad + \sigma\alpha_d k_A^2 \cos(2\theta_J - 2\phi_A)\cos^2\Omega t,
\end{aligned}
\tag{17}
$$

while for driven $p$-wave magnets under CPL and LPL, respectively, we obtain

$$
\begin{aligned}
V_{p,\text{CPL}}^\sigma(t) &= 2Bk_A k \cos(\theta_k - \eta\Omega t) + Bk_A^2 \\
&\quad + \sigma 2\alpha_p k_A \cos(\theta_J - \eta\Omega t),
\end{aligned}
\tag{18}
$$

$$
\begin{aligned}
V_{p,\text{LPL}}^\sigma(t) &= 2Bk_A k \cos(\theta_k - \phi_A)\cos\Omega t + Bk_A^2 \cos^2\Omega t \\
&\quad + \sigma 2\alpha_p k_A \cos(\theta_J - \phi_A)\cos\Omega t,
\end{aligned}
\tag{19}
$$

with

$$k_A = \frac{eE_0}{\hbar\Omega}. \tag{20}$$

At this point, it is important to understand the nature of the elements in Eqs. (16)-(19). For this purpose, we note that, under general conditions, in a Hamiltonian with $k^q$-order terms, Peierls substitution implies that the canonical momentum is of the form as $[k + A(t)]^q \sim \sum_{n=0}^q k^{q-n}[A(t)]^n \sim \sum_{n=0}^q k^{q-n}k_A^n$. Since $q - n \geq 0$ is naturally required, the result of light-matter interaction in the Hamiltonian is expected to include terms proportional to $k_A^n$ with $n = 0, 1, \cdots, q$. As is expected, in the driven unconventional magnet with $q > 0$, there are more non-trivial light-matter interactions related to $\sigma\alpha_q k^{q-n}k_A^n$ and $\theta_J$, $\eta$ or $\phi_A$. Particularly, the terms with $k_A^q$ are momentum-independent, as seen in Eq. (20). Based on this general understanding, one can observe that all the first lines of Eqs. (16-19), proportional to $Bk_A$, originate from the coupling between light and kinetic energy ($\sim Bk^2$), which are expected in a wide range of materials hosting parabolic dispersions and thus considered as trivial. Interestingly, the second and the third lines of Eqs. (16-19) involve the coupling between light, unconventional magnetism, and momentum via $k_A$, $\sigma\alpha_{d,p}$, and $\boldsymbol{k}$, respectively, which unveil a rather non-trivial interaction due to the interplay between light and unconventional magnetism.

In the driven $d$-wave altermagnets, the non-trivial light-matter interaction includes both momentum-dependent term, $\sigma\alpha_d k_A k$ [the second line of Eq. (16) and (17)], and momentum-independent terms, $\sigma\alpha_d k_A^2$ [the third line of Eq. (16) and (17)], both of which depend on the spin via $\sigma$. In the CPL case, these elements are also determined by the angle $\theta_k$, the unconventional magnetic direction $\theta_J$, the left/right handed polarization defined by $\eta$, while for LPL

they are determined by $\theta_k$, the real space polarization direction $\phi_A$, and $\theta_J$. The momentum-dependent light-matter interactions, $\sigma \alpha_d k_A k$, appear with a linear dependence in momentum $k$ and in the magnetic strength $\alpha_d$, which behaves as the Rashba spin-orbit coupling or the $p$-wave magnet field. Moreover, the momentum-independent terms, $\sigma \alpha_d k_A^2$, clearly behave as a Zeeman-like ($s$-wave-magnet-like) effect. We remark that without the drive or altermagnetism, i.e., $k_A = 0$ or $\alpha_d = 0$, these momentum-dependent and momentum-independent contributions vanish, which clearly demonstrates that they originate from the interplay between light and altermagnetism. In the case of $p$-wave magnets, the result of coupling them to the light drive only gives rise to a Zeeman-like effect proportional to $\sigma 2\alpha_p k_A$ [the second line of Eq. (18) and (19)], which occurs due to the linear dependence of momentum in the static $p$-wave magnet Such a light-matter effect is also considered to be non-trivial because it generates a momentum-independent effect related to the unconventional $p$-wave field and the light amplitude, which is also determined by $\theta_J$, $\eta$ or $\phi_A$. As a result, in both $p$- and $d$-wave unconventional magnets, the application of a light drive induces a non-trivial light-matter interaction. It is worth noting that, due the choice of the Néel vector along $z$ axis, the light induced terms preserve the spin direction, but our results above can be easily generalized for any direction [119, 225] or perhaps using a more realistic modelling that involves sublattices [6, 17, 18, 66]. A consequence of modifying the spin texture in momentum space by light could introduce rotating fields that promote a spin precession [129], thus enriching the emergent physics.

We close this part by stressing that light drives induce non-trivial light-matter interactions in unconventional magnets. These interactions are governed by the Fermi surface symmetry of the underlying unconventional magnet, as well as by the specific properties of the driving field. Similar effects are also observed in unconventional magnets with higher angular momentum, characterized here by $q > 2$; see App. A. Driven unconventional magnets thus offer an interesting platform for exploring light-induced spin-dependent phenomena, which, in combination with other phases such as superconductivity, can further lead to states that are absent in equilibrium.

### 3.2 Driven unconventional magnets with conventional superconductivity

As in the previous subsection, we derive the time-dependent Hamiltonian for $d$-wave and $p$-wave unconventional magnets with conventional superconductivity by incorporating the light drives via Peierls substitution in Eqs. (7). We obtain

$$\mathcal{H}_{j,\beta}^{\nu}(\boldsymbol{k}, t) = \mathcal{H}_j^{\nu}(\boldsymbol{k}) + \mathcal{V}_{j,\beta}^{\nu}(t), \tag{21}$$

$j = \{p, d\}$ denotes the type of unconventional magnet, $\mathcal{H}_j^{\nu}(\boldsymbol{k})$ is the static Hamiltonian for an unconventional magnet with superconductivity given by Eqs. (7), and $\mathcal{V}_{j,\beta}^{\nu}(t)$ characterizes the light-induced effect due to the drive denoted by $\beta = \{\text{LPL}, \text{CPL}\}$. Here, for $d$-wave magnets with conventional superconductivity under CPL and LPL, we obtain $\mathcal{V}_{j,\beta}^{\nu}(t)$ given by

$$
\begin{aligned}
\mathcal{V}_{d,\text{CPL}}^{\nu}(t) = {} & 2B k_A k \cos(\theta_k - \eta\Omega t)\tau_0 \\
& + 2\nu\alpha_d k_A k \cos(\theta_k - 2\theta_J + \eta\Omega t)\tau_z \\
& + \nu\alpha_d k_A^2 \cos(2\theta_J - 2\eta\Omega t)\tau_0,
\end{aligned} \tag{22}
$$

$$
\begin{aligned}
\mathcal{V}_{d,\text{LPL}}^{\nu}(t) = {} & 2B k_A k \cos(\theta_k - \eta\Omega t)\tau_0 \\
& + 2\nu\alpha_d k_A k \cos(\theta_k - 2\theta_J + \eta\Omega t)\tau_z \\
& + \nu\alpha_d k_A^2 \cos(2\theta_J - 2\eta\Omega t)\tau_0.
\end{aligned} \tag{23}
$$

Similarly, for $p$-wave magnets with superconductivity under CPL and LPL, respectively, we obtain

$$
\begin{aligned}
\mathcal{V}_{p,\text{CPL}}^{\nu}(t) &= 2Bk_A k \cos\left(\theta_k - \eta\Omega t\right)\tau_0 + Bk_A^2 \tau_z \\
&\quad + \nu 2\alpha_p k_A \cos\left(\theta_J - \eta\Omega t\right)\tau_z,
\end{aligned}
\tag{24}
$$

$$
\begin{aligned}
\mathcal{V}_{p,\text{LPL}}^{\nu}(t) &= 2Bk_A k \cos\left(\theta_k - \phi_A\right)\cos\Omega t\,\tau_0 + Bk_A^2 \cos^2\Omega t\,\tau_z \\
&\quad + 2\nu\alpha_p k_A \cos\left(\theta_p - \phi_A\right)\cos\Omega t\,\tau_z,
\end{aligned}
\tag{25}
$$

At this point, we remark that Eqs. (21)-(25) acquire the same contributions, with the same parameters, as Eq. (15)-(19), including including both trivial and non-trivial light-matter interactions in relation to $Bk_A\tau_0$ and $\nu\alpha_{d,p}k_A$, respectively. Nevertheless, Eqs. (21)-(25) unveil interesting properties in relation to superconductivity. A key effect is that the non-trivial light-matter interactions have different effects on quasielectrons and quasiholes, which is evident by noting the appearance of the Pauli $\tau_z$ matrix since it is in Nambu electron-hole space. This effect appears, for instance, in the momentum-dependent term of the $d$-wave case proportional to $2\nu\alpha_d k_A k\tau_z$, see the second line of Eqs. (22) and (23)]; this term provides a $p$-wave-like spin-momentum coupling as we discussed in the previous subsection. Moreover, in the $p$-wave case, only the momentum-independent term, proportional to $2\nu\alpha_p k_A\tau_z$ develops the opposite light-induced effect on quasielectrons and quasiholes, see the second lines of Eqs. (24) and (25). The features thus suggest a very likely impact on the emergent superconducting properties, such as on the spin-triplet density and Cooper pair symmetries, of driven unconventional magnets with conventional superconductivity.

## 3.3   Floquet description

To further investigate the interplay effect of the time-periodic drive on unconventional magnetism, we employ Floquet theory by mapping the time-dependent system into a time-independent representation in extended frequency (Floquet) space. Floquet theory exploits the periodicity introduced by the time-periodic drives. Thus, since the time-periodic Hamiltonian in Eqs. (15) and (21) are periodic in time, $H_q^{\sigma}(\boldsymbol{k},t) = H_q^{\sigma}(\boldsymbol{k},t+T)$ with period $T = 2\pi/\Omega$, the corresponding solutions of the time-dependent Schrödinger equation

$$
i\hbar\,\partial_t \Psi(\boldsymbol{k},t) = H_q^{\sigma}(\boldsymbol{k},t)\Psi(\boldsymbol{k},t),
\tag{26}
$$

are written as

$$
\Psi(\boldsymbol{k},t) = e^{-i\epsilon t/\hbar}\,\Phi(\boldsymbol{k},t),
\tag{27}
$$

where $\epsilon$ is the quasienergy and $\Phi(\boldsymbol{k},t) = \Phi(\boldsymbol{k},t+T)$ is the time-periodic Floquet mode [128, 226–228]. Substituting into the Schrödinger equation, the Floquet state satisfies the eigenvalue equation

$$
\left[H_q^{\sigma}(\boldsymbol{k},t) - i\hbar\,\partial_t\right]\Phi(\boldsymbol{k},t) = \epsilon\,\Phi(\boldsymbol{k},t).
\tag{28}
$$

We now use the periodicity of $\Phi(\boldsymbol{k},t)$ and expand it in Fourier series as

$$
\Phi(\boldsymbol{k},t) = \sum_{n} e^{-in\Omega t}\Phi_n(\boldsymbol{k}).
\tag{29}
$$

The same is done for the Hamiltonian $H_q^{\sigma}(\boldsymbol{k},t)$. Then, the problem becomes an eigenvalue matrix equation in Floquet space,

$$
H_F^{\sigma}(\boldsymbol{k})\Phi(\boldsymbol{k}) = \epsilon\,\Phi(\boldsymbol{k}),
\tag{30}
$$

where $\Phi(\boldsymbol{k}) = (\cdots, \Phi_{-1}^{\sigma}, \Phi_0^{\sigma}, \Phi_{+1}^{\sigma}, \cdots)^{\mathrm{T}}$ is a vector of Floquet modes, while $H_F^{\sigma}(\boldsymbol{k})$ is the Floquet Hamiltonian and given by

$$
H_F^{\sigma}(\boldsymbol{k}) = \begin{pmatrix}
\ddots & \ddots & \ddots & \ddots & \ddots & \ddots & \ddots \\
\ddots & H_0^{\sigma} + 2\hbar\Omega & H_{+1}^{\sigma} & H_{+2}^{\sigma} & \ddots & \ddots & \ddots \\
\ddots & H_{-1}^{\sigma} & H_0^{\sigma} + \hbar\Omega & H_{+1}^{\sigma} & H_{+2}^{\sigma} & \ddots & \ddots \\
\ddots & H_{-2}^{\sigma} & H_{-1}^{\sigma} & H_0^{\sigma} & H_{+1}^{\sigma} & H_{+2}^{\sigma} & \ddots \\
\ddots & \ddots & H_{-2}^{\sigma} & H_{-1}^{\sigma} & H_0^{\sigma} - \hbar\Omega & H_{+1} & \ddots \\
\ddots & \ddots & \ddots & H_{-2}^{\sigma} & H_{-1}^{\sigma} & H_0^{\sigma} - 2\hbar\Omega & \ddots \\
\ddots & \ddots & \ddots & \ddots & \ddots & \ddots & \ddots
\end{pmatrix}. \tag{31}
$$

We see that the Floquet Hamiltonian $H_F^{\sigma}(\boldsymbol{k})$ has infinite dimension, with time-independent components akin to a tight-binding lattice in frequency space. Here, the matrix elements are defined as

$$
H_n^{q,\sigma}(\boldsymbol{k}) = \frac{1}{T} \int_0^T dt \, H_q^{\sigma}(\boldsymbol{k}, t) \, e^{in\Omega t}, \tag{32}
$$

describe processes involving absorption ($n > 0$) or emission ($n < 0$) of $|n|$ photons, coupling Floquet states $\Phi_m$ to $\Phi_{m+n}$ [226] and $H_{-n}^{q,\sigma}(\boldsymbol{k})$ are obtained through $H_{-n}^{q,\sigma}(\boldsymbol{k}) = [H_n^{q,\sigma}(\boldsymbol{k})]^{\dagger}$. The diagonal elements of $H_F^{\sigma}(\boldsymbol{k})$ are shifted by $n\hbar\Omega$ and are coupled by the drive via the emission and absorption of one (two) photons.

The Floquet approach discussed above applies to unconventional magnets without and with superconductivity, whose Floquet Hamiltonian we denote, respectively, as $H_F^{q,\sigma}(\boldsymbol{k})$ and $\mathcal{H}_F^{q,\nu}(\boldsymbol{k})$. In the following, we detail the explicit forms of the Floquet components $H_{n,\beta}^{q,\sigma}(\boldsymbol{k})$ and $\mathcal{H}_{n,\beta}^{q,\nu}(\boldsymbol{k})$ under light drives denoted by $\beta = \{\text{CPL}, \text{LPL}\}$, focusing specifically on $d$-wave and $p$-wave unconventional magnets without and with superconductivity. For $d$-wave altermagnets in the normal state, we find that the elements of the Floquet Hamiltonian are given by

$$
\begin{aligned}
H_{0,\beta}^{d,\sigma} &= \begin{cases} H_d^{\sigma}(\boldsymbol{k}) + Bk_A^2, & \text{for } \beta = \text{CPL}, \\ H_d^{\sigma}(\boldsymbol{k}) + \frac{1}{2}k_A^2[B + \sigma\alpha_d \cos(2\theta_J - 2\phi_A)], & \text{for } \beta = \text{LPL}, \end{cases} \\[2mm]
H_{+1,\beta}^{d,\sigma} &= k_A k \times \begin{cases} Be^{i\theta_k} + \sigma\alpha_d e^{i(2\theta_J - \theta_k)}, & \text{for } \beta = \text{CPL}, \\ B\cos(\theta_k - \phi_A) + \sigma\alpha_d \cos(\theta_k - 2\theta_J + \phi_A), & \text{for } \beta = \text{LPL}, \end{cases} \\[2mm]
H_{+2,\beta}^{d,\sigma} &= \frac{1}{4}k_A^2 \times \begin{cases} \sigma 2\alpha_d e^{2i\theta_J}, & \text{for } \beta = \text{CPL}, \\ B + \sigma\alpha_d \cos(2\theta_J - 2\phi_A), & \text{for } \beta = \text{LPL}, \end{cases}
\end{aligned} \tag{33}
$$

while for $p$-wave normal state magnets we obtain

$$
\begin{aligned}
H_{0,\beta}^{p,\sigma} &= \begin{cases} H_p^{\sigma}(\boldsymbol{k}) + Bk_A^2, & \text{for } \beta = \text{CPL}, \\ H_p^{\sigma}(\boldsymbol{k}) + Bk_A^2/2, & \text{for } \beta = \text{LPL}, \end{cases} \\[2mm]
H_{+1,\beta}^{p,\sigma} &= k_A \times \begin{cases} kBe^{i\theta_k} + \sigma\alpha_p e^{i\theta_J}, & \text{for } \beta = \text{CPL}, \\ kB\cos(\theta_k - \phi_A) + \sigma\alpha_p \cos(\theta_J - \phi_A), & \text{for } \beta = \text{LPL}, \end{cases} \\[2mm]
H_{+2,\beta}^{p,\sigma} &= \begin{cases} 0, & \text{for } \beta = \text{CPL}, \\ Bk_A^2/4, & \text{for } \beta = \text{LPL}. \end{cases}
\end{aligned} \tag{34}
$$

For $d$-wave altermagnets with conventional superconductivity, we obtain

$$\mathcal{H}_{0,\beta}^{d,\nu} = \begin{cases} \mathcal{H}_d^\nu(\boldsymbol{k}) + Bk_A^2\tau_z, & \text{for } \beta = \text{CPL}, \\ \mathcal{H}_d^\nu(\boldsymbol{k}) + \frac{1}{2}k_A^2[B\tau_z + \nu\alpha_d\cos(2\theta_J - 2\phi_A)\tau_0], & \text{for } \beta = \text{LPL} \end{cases}$$

$$\mathcal{H}_{+1,\beta}^{d,\nu} = k_A k \times \begin{cases} Be^{i\theta_k}\tau_0 + \nu\alpha_d e^{i(2\theta_J - \theta_k)}\tau_z, & \text{for } \beta = \text{CPL}, \\ B\cos(\theta_k - \phi_A)\tau_0 + \nu\alpha_d\cos(\theta_k - 2\theta_J + \phi_A)\tau_z, & \text{for } \beta = \text{LPL} \end{cases} \quad (35)$$

$$\mathcal{H}_{+2}^{d,\nu} = \frac{1}{4}k_A^2 \times \begin{cases} \nu 2\alpha_d e^{2i\theta_J}\tau_0, & \text{for } \beta = \text{CPL}, \\ B\tau_z + \nu\alpha_d\cos(2\theta_J - 2\phi_A)\tau_0, & \text{for } \beta = \text{LPL}, \end{cases}$$

and, for $p$-wave magnets with conventional superconductivity, we find,

$$\mathcal{H}_{0,\beta}^{p,\nu} = \begin{cases} \mathcal{H}_p^\nu(\boldsymbol{k}) + Bk_A^2\tau_z, & \text{for } \beta = \text{CPL}, \\ \mathcal{H}_p^\nu(\boldsymbol{k}) + Bk_A^2/2\,\tau_z, & \text{for } \beta = \text{LPL}, \end{cases}$$

$$\mathcal{H}_{+1,\beta}^{p,\nu} = k_A \times \begin{cases} kBe^{i\theta_k}\tau_0 + \nu\alpha_p e^{i\theta_J}\tau_0, & \text{for } \beta = \text{CPL}, \\ kB\cos(\theta_k - \phi_A)\tau_0 + \nu\alpha_p\cos(\theta_J - \phi_A)\tau_0, & \text{for } \beta = \text{LPL}, \end{cases} \quad (36)$$

$$\mathcal{H}_{+2}^{p,\nu} = \begin{cases} 0, & \text{for } \beta = \text{CPL}, \\ Bk_A^2/4\,\tau_z, & \text{for } \beta = \text{LPL}. \end{cases}$$

We remind that $H_{-1(-2),\beta}^{d,\sigma}$ and $\mathcal{H}_{-1(-2),\beta}^{p,\sigma}$ are obtained through $H_{-n}^{d,\sigma}(\boldsymbol{k}) = [H_n^{d,\sigma}(\boldsymbol{k})]^\dagger$ and $\mathcal{H}_{-n}^{p,\sigma}(\boldsymbol{k}) = [\mathcal{H}_n^{p,\sigma}(\boldsymbol{k})]^\dagger$, respectively.

Before going further, it is useful to stress some properties of the Floquet elements given Eqs. (33)-(36). While $H_{0,\beta}^{j,\sigma}$ and $\mathcal{H}_{0,\beta}^{j,\nu}$ involve zero photon processes, $H_{+1(+2),\beta}^{j,\sigma}$ and $\mathcal{H}_{+1(+2),\beta}^{j,\nu}$ involve one (two) photon processes. These elements exhibit a rich functionality depending on the type of drive and unconventional magnet. In the case of $n = 0$ Floquet terms, we first note that there appears a uniform chemical potential shift given by $Bk_A^2$ under CPL and $Bk_A^2/2$ under LPL of the normal and superconducting regimes, see $H_{0,\beta}^{j,\sigma}$ and $\mathcal{H}_{0,\beta}^{j,\nu}$ in Eqs. (33)-(36). This is often referred to as the self-doping effect [226], which originates from the light-induced renormalization of the kinetic energy and is hence considered to be trivial. Another feature of the $n = 0$ Floquet terms is that the LPL drive induces an effective Zeeman-like field in the normal state of $d$-wave altermagnets, which is, however, absent in $p$-wave magnets; see e.g., $H_{0,\beta}^{d,\sigma}$ for $\beta = \text{LPL}$ in Eqs. (33) and Eqs. (34). These Zeeman-like fields characterize momentum-independent spin splittings arising from the interplay among the light intensity $k_A^2$, the $d$-wave altermagnetic strength $\alpha_d$, and the angular mismatch between the magnetic orientation $\theta_J$ and the polarization angle $\phi_A$. This emergent spin-dependent field is finite only when the polarization direction deviates from the spin-degenerate axes, i.e., when $\phi_A - \theta_J \neq (2n + 1)\pi/4$ for $n \in \mathbb{Z}$. This unveils that, while $d$-wave altermagnets intrinsically have zero net magnetization, light is able to induce a Zeeman-like field that is likely to affect the magnetization and is sensitive to $\alpha_d$ and $\theta_J$. This effect could have an interesting consequence as it offers a way to probe the underlying $d$-wave altermagnetic order and provides a practical route to probe its strength and orientation [75]. We have also verified that similar Zeeman-like fields appear in unconventional magnets with higher angular momentum, see App. A for a detailed discussion.

For the terms involving one photon $n = +1$, the $d$-wave systems without and with superconductivity develop linearly momentum-dependent terms inherited from the time-dependent Hamiltonian. This enables a photon-induced $p$-wave-like field by combining $d$-wave altermagnetism and light by absorbing or emitting one photon. In the LPL case, this $p$-wave-like term depends on the angle between the $d$-wave magnet orientation $\theta_J$ and the linearly polarized direction $\phi_A$, which provides a tuning knob to control the single-photon process. In relation

to the one-photon terms ($n = +1$) in the driven $p$-wave case, momentum-independent contributions emerge without and with superconductivity, see $H^{p,\sigma}_{+1,\beta}$ and $\mathcal{H}^{p,\nu}_{+1,\beta}$ in Eqs. (33)-(36). Similar to $d$-wave systems under LPL, the $H^{p,\sigma}_{+1,LPL}$ and $\mathcal{H}^{p,\nu}_{+1,LPL}$ in driven $p$-wave magnets are determined by the direction between the $p$-wave orientation $\theta_J$ and the linear polarization $\phi_A$. When it comes to two-photon processes, effects combining driving field and magnetism only appear in $d$-wave altermagnets, see the $n = 2$ sectors in Eqs. (33)-(36).

Moreover, all Floquet components with $|n| \geq 3$ vanish in driven $d$- and $p$-wave unconventional magnets, which is a direct consequence of the momentum order $q$ of the static Hamiltonian. In a system where the static Hamiltonian contains terms up to order $k^q$, the Peierls substitution modifies the canonical momentum as $\boldsymbol{k} \to \boldsymbol{k} + e/\hbar \boldsymbol{A}(t)$, resulting in an expansion of the form $\sum_{n=0}^{q} k^{q-n} k_A^n$. This expansion inherently restricts the light-matter interaction terms to powers $k_A^n$ with $n = 0, 1, \ldots, q$, as the requirement $q - n \geq 0$ must be satisfied. Since the Floquet components $H_n^\sigma(\boldsymbol{k})$ and $\mathcal{H}_n^\sigma(\boldsymbol{k})$ encode processes involving the absorption or emission of $n$ photons, only harmonics up to order $n \leq q$ appear in the driven system. Consequently, all higher-order photon processes with $|n| > q$ are strictly forbidden (see App. A), as the corresponding powers of $k_A$ are absent in the expansion of the light–matter interaction [187].

We close this section by noting that Floquet theory offers a useful ground to study the effect of time-periodic light drives on unconventional magnets, where the inherent interplay between light and unconventional magnetism is fully captured. With this at hand, we are ready to explore how light affects the spin density and superconducting correlations in unconventional magnets. For computational purposes, in the following sections we truncate the Floquet Hamiltonian [Eq.(31)] and consider 11 Floquet sidebands with $n \in [-5, 5]$; we verified that this consideration does not affect the main messages of our work.

## 4 Light-induced Floquet spin density in the normal state

We now use the Floquet Hamiltonian [Eq. (31)] discussed in the previous section and inspect the emergence of spin density in driven unconventional magnets without superconductivity. We obtain the Floquet spin density along $z$ as

$$S_{F,z}(\omega, \boldsymbol{k}) = -\frac{1}{\pi} \text{ImTr}[\sigma_z G_F(\omega + i0^+, \boldsymbol{k})], \tag{37}$$

where

$$G_F(z, \boldsymbol{k}) = [z - H_F(\boldsymbol{k})]^{-1}, \tag{38}$$

is the Floquet Green's function associated to the Floquet Hamiltonian $H_F = \text{diag}(H_F^+, H_F^-)$, with $H_F^+$ defined in Eq. (31) with $z = \omega + i0^+$ and $\sigma_z$ is the thrid Pauli matrix in Floquet space. Using the Floquet components in Sec. 3.3, the Floquet spin density $S_{F,z}(\omega, \boldsymbol{k})$ is obtained in driven unconventional magnets.

To visualize the Floquet spin density, in Fig. 3 we plot $S_{F,z}(\omega, \boldsymbol{k})$ as a function of $k_x$ and $k_y$ for $d_{x^2-y^2}$-wave altermagnets and $p_x$-wave magnets under CPL and LPL. In the case of $d$-wave altermagnets [Fig. 3(a,c)] under CPL and LPL, $S_{F,z}(\omega, \boldsymbol{k})$ develops large values over a series of concentric vertical and horizontal ellipses with positive and negative spin density values, respectively. In the case of $p_x$-wave magnets, the spin density forms a series of circles shifted to positive and negative $k_x$, acquiring negative and positive spin density values, respectively; see Fig. 3(b,d). This implies that the vertical and horizontal ellipses (shifted circles) are spin polarized, akin to what happens with the spin density in the static regime of unconventional magnets [Fig. 2]. The series of ellipses (circles) represent Floquet replicas and arise from the formation of Floquet sidebands, tied to the diagonal elements of the Floquet Hamiltonian

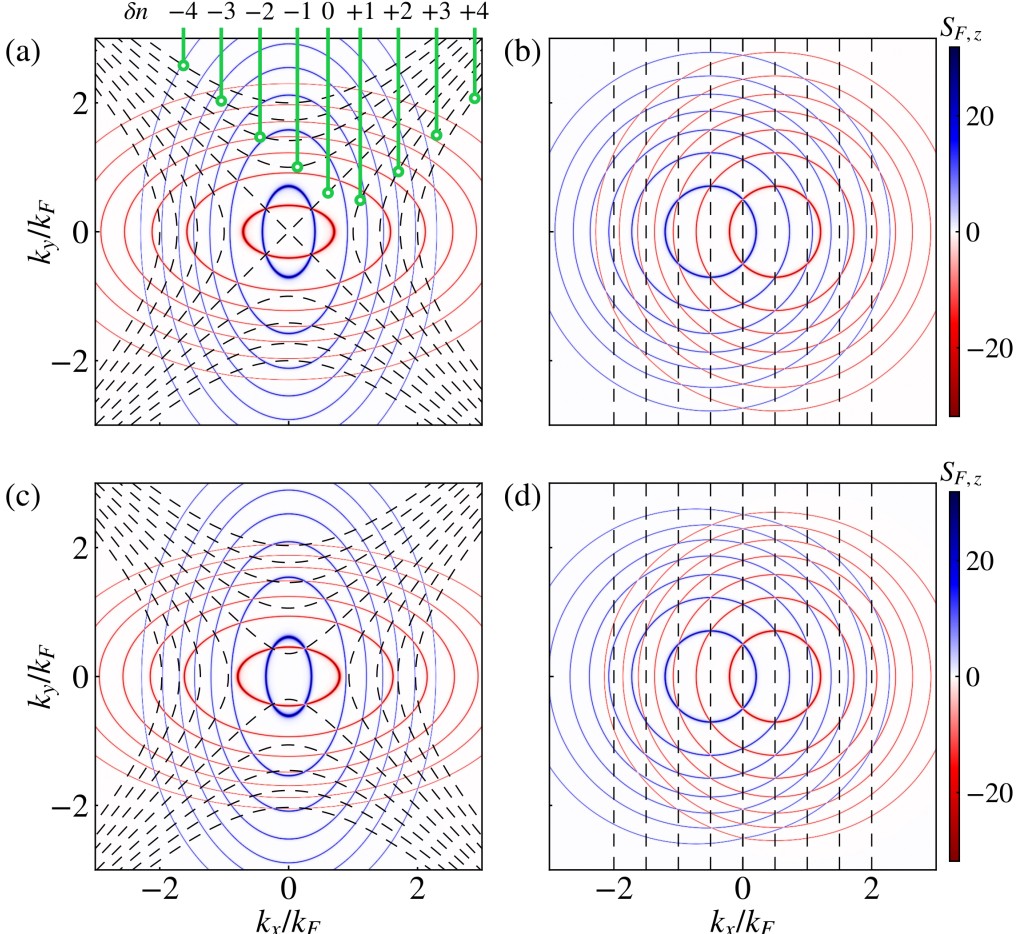

Figure 3: (a) Floquet spin density $S_{\mathrm{F,z}}$ as a function of momenta for $d_{x^2-y^2}$-wave altermagnets under CPL with $\eta = +1$. (b) The same as in (a) but for a $p_x$-wave magnet. (c,d) The same as in (a,b) but under LPL with $\phi_A = 0$. The black dashed lines in (a) connect the spin-degenerate nodes between the $n^{\mathrm{th}}$ and the $m^{\mathrm{th}}$ Floquet sidebands and $\delta n = n - m$ [Eq. (42) and (43)]. Parameters: $B = 1$, $\alpha_{d,p} = 0.5$, $\theta_J = 0$, $\mu = 1$, $\Omega/\mu = 1$, $k_A/k_F = 0.5$, and $k_F = \sqrt{\mu/B} = 1$; 11 Floquet sidebands are considered with $n \in [-5, 5]$.

shifted by $n\Omega$ [Eq. (31)]. It is important to remark some differences between CPL- and LPL-driven cases. For instance, for the $d$-wave altermagnet under LPL, the size of the ellipses with the negative spin density is larger, while it is smaller for the positive spin density; this originates from the contribution of the Zeeman-like effect of the light drive, i.e. $\sigma \frac{1}{2} k_A^2 \alpha_d \cos(2\theta_J - 2\phi_A)$ in the $n = 0$ sector Hamiltonian ($H_{0,\beta}^{d,\sigma}$) in Eqs. (33). In contrast to the $d$-wave cases, for $p$-wave magnets under CPL and LPL, the size of the circular spin densities is the same, see $H_{0,\beta}^{p,\sigma}$ in Eqs. (34). Furthermore, another difference between the effects of CPL and LPL is that, for the $n = 0$ Floquet sector of the $d$- and $p$-wave unconventional magnets driven by CPL, the spin density develops finite values within the ellipses and circles, respectively, unlike the vanishing values for LPL-driven cases. It is also worth noting that the presence of Floquet replicas leads to a larger number of spin-degenerate nodes due to additional intersections between opposite-spin Fermi surfaces associated with different Floquet indices, as indicated by overlaid dashed curves in [Fig. 3(a)]. We remind that, in the static case, the $d$-wave ($p$-wave) unconventional magnets host four (two) spin-degenerate nodes in momentum space, as shown in Fig. 2(b,d).

To understand the origin of spin-degenerate nodes in the Floquet spin density discussed above, we consider an effective model involving two arbitrary Floquet sidebands (2FSB) with opposite spin orientations. The reduced Floquet Hamiltonian in the subspace spanned by the two sidebands $\left(\Phi_n^\sigma(\boldsymbol{k}), \Phi_m^{-\sigma}(\boldsymbol{k})\right)^{\mathrm{T}}$ is given by

$$H_{2\mathrm{FSB}} \approx \begin{pmatrix} H_0^\sigma(\boldsymbol{k}) + n\hbar\Omega & 0 \\ 0 & H_0^{-\sigma}(\boldsymbol{k}) + m\hbar\Omega \end{pmatrix}, \tag{39}$$

where the absence of off-diagonal terms reflects the lack of spin-mixing terms in the unconventional magnets, see Eq. (1). In principle, identifying spin-degenerate nodes between opposite-spin Floquet sidebands would require solving the full Floquet Hamiltonian in the infinite Floquet space [Eq. (31)]. To overcome this difficulty and to provide a transparent analytical description of the momentum-space distribution of these nodes, we extract two opposite-spin diagonal blocks of the full Floquet Hamiltonian [Eq. (31)] and neglect the off-diagonal Floquet couplings, such as $H_{\pm 1}$ and $H_{\pm 2}$. This procedure yields an approximate effective description of two opposite-spin Floquet sidebands, which is given by Eq. (39). Despite this approximation, Eq. (39) accurately captures the locations of the spin-degenerate nodes between arbitrary opposite-spin Floquet sidebands, as shown below.

To find the condition for spin degeneracy between Floquet sidebands ($\Phi_n^\sigma$ and $\Phi_m^{-\sigma}$), we set the diagonal to be equal and obtain

$$\sigma J_{d,p}(\boldsymbol{k}) + \sigma M(k_A, \phi_A) + \delta n\,\hbar\Omega/2 = 0, \tag{40}$$

where $\delta n = (n - m)$ and $M(k_A, \phi_A)$ is a light-induced Zeeman-like term. $M(k_A, \phi_A)$ is defined as

$$M(k_A, \phi_A) = \begin{cases} \frac{1}{2}\alpha_d k_A^2 \cos(2\theta_J - 2\phi_A), & \text{for LPL-driven } d\text{-wave altermagnets,} \\ 0, & \text{other cases,} \end{cases} \tag{41}$$

and appears only under LPL in $d$-wave systems.

For a $d_{x^2-y^2}$-wave altermagnet with $\theta_J = 0$, Eq. (40) reduces to

$$\sigma \alpha_d(k_x^2 - k_y^2) + \sigma M(k_A, \phi_A) = \delta n\,\hbar\Omega/2, \tag{42}$$

which describes a family of spin-degenerate parabolas in the $k_x$–$k_y$ plane, see Figs. 3(a,c). When $M(k_A, \phi_A) = 0$, as in a $d$-wave altermagnet under CPL, for $\delta n = 0$, the spin degeneracy occurs along the nodal lines $k_y = \pm k_x$, serving as asymptotes for these parabolas. By contrast, for a driven $p_x$-wave magnet with $\theta_J = 0$, where the magnetic order is linear in momentum, the degeneracy condition simplifies to

$$\alpha_p k_x = -\delta n\,\hbar\Omega, \tag{43}$$

corresponding to a series of equally spaced vertical lines in momentum space, separated by $\hbar\Omega/\alpha_p$, see Figs. 3(b,d).

We can thus conclude that unconventional magnets under time-periodic drives acquire a Floquet spin-triplet density due to Floquet sidebands, exhibiting properties that can help identify the type of unconventional magnetism. We verified that these results are valid for strong $d$-wave altermagnets, see Appendix B.

## 4.1 Floquet spin density projected onto the zero-photon subspace

To further investigate the influence of higher-order Floquet components $H_{n\neq 0}^\sigma(\boldsymbol{k})$, it is instructive to project the Floquet spin density onto the zero-photon states. The projected Floquet spin density is defined as

$$S_z^{(0)}(\omega, \boldsymbol{k}) = -\frac{1}{\pi}\mathrm{Im}\,\mathrm{Tr}\left[\sigma_z P^\dagger G_F(\omega + i0^+) P\right], \tag{44}$$

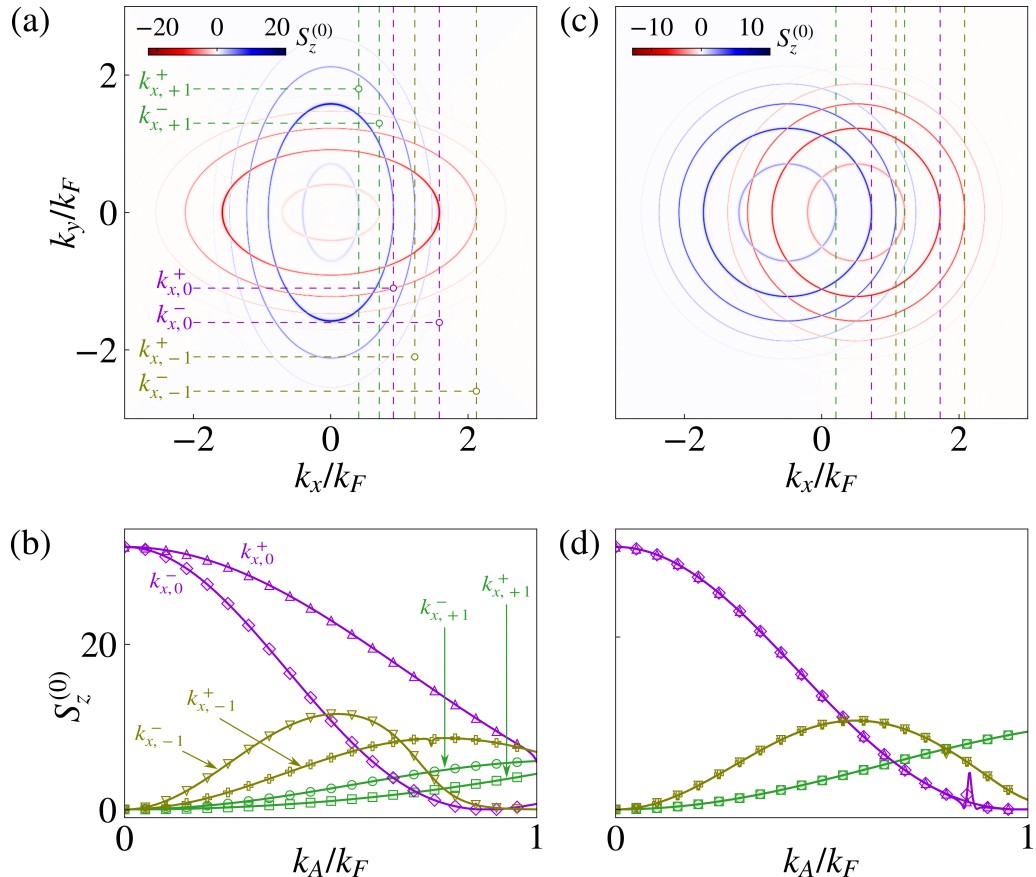

Figure 4: (a,c) Floquet spin density projected to the zero-photon subspace ($S_z^{(0)}$) for $d_{x^2-y^2}$-wave altermagnets (a) and $p_x$-wave magnets, both under CPL. (b,d) $S_z^{(0)}$ as a function of the driving amplitude $k_A$ for the cases shown in (a,b) at $k_y = 0$ and $k_x$ indicated in (a); $k_x$ is given by Eq. (45) for (b), while by Eq. (46) for (d). Parameters as in Fig. 4.

where $P = (\cdots, 0, \Phi_0(\boldsymbol{k}), 0, \cdots)^{\mathrm{T}}$ is a projector onto the $n = 0$ Floquet sector, the Green's function $G_F$ is defined in Eq. (38). This formulation enables us to isolate the contribution of photon-assisted processes, including absorption and emission, thereby revealing the impact of inter-sideband coupling encoded in the off-diagonal Floquet terms. The projected spin density $S_z^{(0)}(\omega, \boldsymbol{k})$ for $d$- and $p$-wave unconventional magnets under CPL is shown in Fig. 4(a,c). While qualitatively similar to the full Floquet spin density presented in Fig. 3(a), $S_z^{(0)}(\omega, \boldsymbol{k})$ in Fig. 4(a,c) primarily captures contributions from the zeroth-order Floquet sector $H_0^\sigma(\boldsymbol{k})$ and it also possesses information about the coupling between nearest-neighbor and next-nearest-neighbor sidebands, determined by $H_{\pm 1}^\sigma(\boldsymbol{k})$ and $H_{\pm 2}^\sigma(\boldsymbol{k})$, respectively. To visualize the effect of these one- and two-photon absorption/emission processes, we note that $S_z^{(0)}(\omega, \boldsymbol{k})$ develops positive and negative values, which can be interpreted as peaks and dips, respectively. In this regard, for $d$-wave altermagnets under CPL, one can estimate the momenta at which the peaks ($S_z^{(0)} > 0$) and dips ($S_z^{(0)} < 0$) happen: at $k_y = 0$, the peaks/dips happen at

$$k_{x,n}^\sigma = \pm\sqrt{\frac{\omega - n\Omega}{B + \sigma\alpha_d}}, \tag{45}$$

where $n = 0, \pm 1, \pm 2$ labels the Floquet sidebands and $\sigma = \pm$; the peak is obtained for $\sigma = +$, while the dip for $\sigma = -$. The $k_{x,n}^\sigma$ in Eq. (45) are obtained by solving $H_0^\sigma(\boldsymbol{k}) + n\hbar\Omega = 0$ in

Eq. (39). In Fig. 4(b,d), we plot the $S_z^{(0)}(\omega, \boldsymbol{k})$ as a function of $k_A$ at $k_y = 0$ and $k_x$ given by Eq. (45) for $d$-wave altermagnets under CPL. As the driving amplitude $k_A$ increases, the weight at $k_{x,0}^{\sigma}$ is suppressed in a spin-dependent manner due to the magnetic character of the two-photon Floquet terms $H_{\pm 2}^{\sigma}(\boldsymbol{k})$. Simultaneously, the peaks at $k_{x,\pm 1}^{\sigma}$ become dominant, overtaking the central peak at strong driving as the effect $H_{\pm 1}^{\sigma}(\boldsymbol{k})$ becomes significant [Fig. 4(b)]. Similar behavior is observed in the CPL-driven $p$-wave magnet case shown in Figs. 4(c)–(d). In this case, the projected spin density peaks/dips are located at

$$k_{x,n}^{\sigma} = \pm\sqrt{\frac{\omega + \mu - n\Omega}{B}} + \sigma\alpha_p, \tag{46}$$

reflecting the linear $p$-wave dispersion. Unlike the $d$-wave case, the absence of $H_{\pm 2}^{\sigma}(\boldsymbol{k})$ terms in the $p$-wave magnet under light eliminates the spin-dependent suppression, yet the one-photon Floquet replicas $k_{x,\pm 1}^{\sigma}$ still dominate in the strong driving regime, as seen in Fig. 4(d).

Compared to CPL-driven unconventional magnets, LPL introduces anisotropic driving effects that depend on the polarization direction $\phi_A$. The peaks and dips of the projected spin density for $k_y = 0$ occur at

$$k_{x,n}^{\sigma} = \pm\sqrt{\frac{\omega + \mu + \sigma M(k_A, \phi_A) - n\Omega}{B + \sigma\alpha_d}}, \tag{47}$$

where $M(k_A, \phi_A)$ [Eq. (41)] is the effective light-induced Zeeman field introduced by the LPL drive. In particular, when $\phi_A = \pi/2$, the driving field becomes orthogonal to the $k_x$ axis, resulting in a suppression of $H_{\pm 1}^{\sigma}$ and thereby reducing the sideband-induced features. This anisotropy is also evident in the LPL-driven $p$-wave magnet. Although no light-induced Zeeman field appears in this case, the amplitude of $S_{0,z}$ at $k_{x,\pm 1}^{\sigma}$ is modulated as $\cos\phi_A$ and vanishes at $\phi_A = \pi/2$, consistent with $H_{\pm 1}^{\sigma} \sim \cos\phi_A$ [Eq. (34)].

In conclusion, the Floquet spin densities and their zero-photon projections reveal the roles of different Floquet components. The diagonal components $H_0^{\sigma}(\boldsymbol{k})$ give rise to multiple spin-degenerate nodes in momentum space, modifying the magnetic structure beyond that of the static unconventional magnet. Meanwhile, the off-diagonal components $H_{n\neq 0}^{\sigma}(\boldsymbol{k})$ encode the photon-assisted processes that dress the quasiparticle states. Both the Floquet spin density and its projection to the zero-photon subspace unveil properties of the underlying unconventional magnet. We have also verified that the same is true for the spin density in driven unconventional magnets with conventional superconductivity and a $d$-wave magnet with a strong exchange field, see Appendix B.

# 5 Light-induced Floquet spin-triplet Cooper pairs

In this part, we inspect the Floquet BdG spectrum and the emergence of spin-triplet Cooper pairs in unconventional magnets with spin-singlet $s$-wave superconductivity subjected to time-periodic drives.

## 5.1 Floquet BdG spectrum

To obtain the Floquet BdG spectrum, we diagonalize the Floquet Hamiltonian $\mathcal{H}_F^{\gamma}(\boldsymbol{k})$ discussed in Sec. 3.3 and given by Eq. (31), whose matrix elements are given by Eqs. (35) and (36) for $d$- and $p$-wave unconventional magnets. In Fig. 5(a,b), we present the Floquet BdG spectrum as a function of $k_x$ for $d$- and $p$-wave unconventional magnets with spin-singlet $s$-wave super-conductivity under CPL. Compared with the static regime in Fig. 2(e,g), the Floquet spectra in Fig. 5(a,b) support multiple bands which then hybridize and develop gaps. This originates

from the fact that Bogoliubov quasiparticles are composed of electrons in the $n$-photon state pairing with holes in the $m$-photon states mediated by absorption or emission of $|n-m|$ photons.

To better understand the induced gaps between Floquet sidebands, we consider an effective model involving two sidebands in Nambu space $\left(\Phi_n^\nu(\boldsymbol{k}), \Phi_m^{-\nu,\dagger}(\boldsymbol{k})\right)^{\mathrm{T}}$. The two Floquet sideband model (2FSB) is given by

$$\mathcal{H}_{2\mathrm{FSB}} \approx \begin{pmatrix} H_q^\nu(\boldsymbol{k}) + n\hbar\Omega & \Delta_{n,m}^\nu \\ \Delta_{n,m}^{\nu,\dagger} & -[H_q^{-\nu}(-\boldsymbol{k})]^* + m\hbar\Omega \end{pmatrix}, \tag{48}$$

where the index $\nu = +$ ($\nu = -$). Given by basis of Eq. (48), it describes Bogoliubov quasiparticles formed by spin-up (spin-down) electrons in the $n^{\mathrm{th}}$ Floquet sideband and spin-down (spin-up) holes in the $m^{\mathrm{th}}$ Floquet sideband by emitting/absorbing $|n-m|$ photons. Moreover, $\Delta_{n,m}$ in Eq. (48) characterizes the multiple superconducting gaps between Floquet sidebands, in addition to the static pair potential $2\Delta_{0,0} = \Delta$. We observe that gaps between Floquet bands ($\Delta_{n,m}$) decay with increasing photon number difference $|n-m|$, consistent with the perturbative nature of higher-order Floquet processes. The behavior of the multiple gaps can be seen in Fig. 5(a) for a $d_{x^2-y^2}$-wave altermagnet with conventional superconductivity under CPL. By equating the quasielectron and quasihole bands in the $n$- and $m$-photon sectors from the diagonal terms in Eq. (48), respectively, the momenta at which the gap opens can be found. In particular, for $k_y = 0$, the induced gaps open at

$$k_{x,\delta n} = \pm\sqrt{\frac{\mu - (\delta n)\hbar\Omega/2}{B}}, \tag{49}$$

with $\delta n = n - m$. The energy at these intersection points defines the gap center energies at $k_y = 0$,

$$E_{c,n,\delta n}^\nu = (B + \nu\alpha_d)k_{x,\delta n}^2 - \mu + n\hbar\Omega, \tag{50}$$

which are spin-dependent due to the unconventional magnetic term $\alpha_d$. At this point, it is also worth noting that, in the LPL-driven case, we verified that the gap centers acquire additional spin-dependent shifts via the effective Zeeman-like field $M(k_A, \phi_A)$ as a result of the light–matter interaction [Eq. (40)]. For $d_{xy}$-wave altermagnets with $\theta_J = \pi/4$, the unconventional magnetic field vanishes along the $k_x$ axis [see Eq. (4)]. This results in a spin-degenerate Floquet BdG spectrum in the $E$–$k_x$ plane, which is similar to the driven $s$-wave superconductors [187]. Thus, the spin dependence in Eq. (50) vanishes and the multiple band gaps are symmetrically located around energies $E = (n+m)\hbar\Omega/2$, governed purely by the photon energy difference between two paired Floquet sidebands, which can be obtained from Eqs. (50) with $\alpha_d = 0$.

In the CPL-driven $p_x$-wave magnet with conventional superconductivity, multiple superconducting gaps also emerge in the Floquet BdG spectrum, as shown in Fig. 5(b). Unlike the $d$-wave case, the center of each superconducting gap appears at energy $E = \delta n\hbar\Omega$, where $\delta n = n - m$ denotes the difference in photon numbers between paired electron and hole states. However, the gap-opening momenta are spin-split and, for $k_y = 0$, are located at

$$k_{x,\delta n}^\nu = -\nu\frac{\alpha_p}{B} + \sqrt{\left(\frac{\alpha_p}{B}\right)^2 + \frac{\mu - (\delta n)\hbar\Omega}{B}}, \tag{51}$$

where $\nu = \pm$ denotes the spin orientation and the shift originates from $\alpha_p$. For $p_y$-wave magnet with $\theta_J = \pi$, the magnetic effect vanishes in $E$–$k_x$ plane [see Eq. (4)], which results in a spin-degenerate Floquet BdG spectrum similar to the driven $s$-wave superconductor [187] and $d_{xy}$-wave altermagnets with $\theta_J = \pi/4$ along $k_x$ mentioned above. Thus, the spin-dependence of

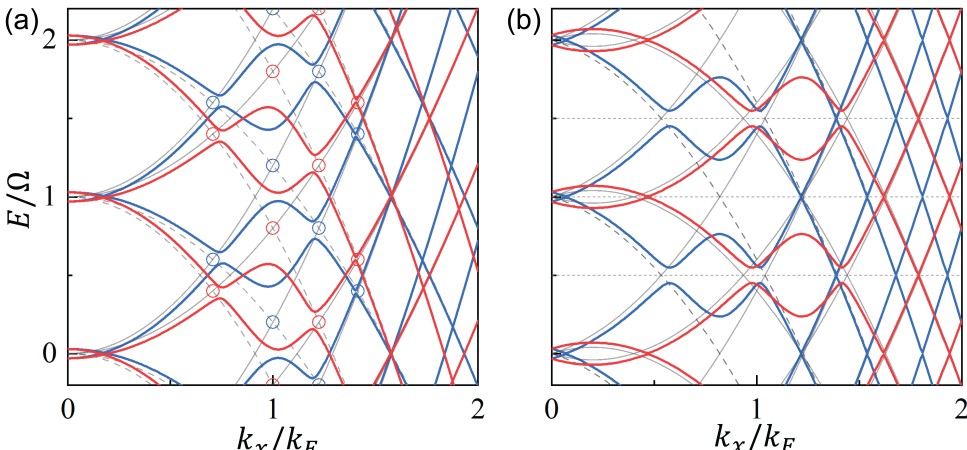

Figure 5: (a) Floquet BdG spectrum of a CPL-driven $d_{x^2-y^2}$-wave magnet with $s$-wave superconductivity with $\eta = +1$ and $\theta_J = 0$. The centers of the superconducting gaps, defined by Eq. (50), are marked by blue and red circles corresponding to $\nu = +$ and $\nu = -$, respectively. (b) Floquet BdG spectrum of a CPL-driven $p_x$-wave magnet with $s$-wave superconductivity. Thin gray solid (dashed) lines represent electron (hole) branches. In both panels, the driving amplitude is $ak_A = 0.5$ and the photon energy is $\hbar\Omega = 1$. All other magnetic and superconducting parameters are identical to those used in Fig. 2; 11 Floquet sidebands are considered with $n \in [-5, 5]$.

the gap-opening momenta vanishes, which can be obtained from Eq. (51) with $\alpha_p = 0$. We also verified that the effect of the multiple superconducting gaps in the CPL-driven $p$-wave magnet is maintained in the LPL-driven cases, since the interpretation of these gaps is related to the $n = 0$ sector of the Floquet Hamiltonian [Eqs. (36)], which is shared by both the CPL and LPL cases except for the shift chemical potential.

We thus conclude this part by noting that Floquet unconventional magnets with conventional superconductivity develop a Floquet spectrum with gaps due to photon-assisted coupling between quasielectron and quasihole states in different Floquet sidebands. This situation thus suggests the emergence of superconducting correlations between Floquet sidebands, which we address in the next subsection.

## 5.2 Floquet Cooper pairs

We now focus on the formation of Cooper pairs between Floquet sidebands in Floquet unconventional magnets with spin-singlet $s$-wave superconductivity modelled by Eq. (31) in Sec. 3.3. For this purpose, we characterize Cooper pairs by using the anomalous Green's function, which is obtained from the electron-hole component of the total Floquet Green's function in Nambu space associated with the Floquet Hamiltonian [Eq. (31)].

For a pedagogical purpose, we start by defining the anomalous Green's function [195, 196, 199],

$$F^{\sigma_1,\sigma_2}(\boldsymbol{k}_1, \boldsymbol{k}_2; t_1, t_2) = -i \left\langle \hat{\mathcal{T}} C_{\boldsymbol{k}_1,\sigma_1}(t_1) C_{\boldsymbol{k}_2,\sigma_2}(t_2) \right\rangle, \tag{52}$$

where $\hat{\mathcal{T}}$ denotes the time-ordering operator, and $C_{\boldsymbol{k},\sigma}(t)$ is the annihilation operator for an electron with momentum $\boldsymbol{k}$ and spin $\sigma$ at time $t$. Since the paired quasiparticles are fermions, Fermi-Dirac statistics imposes the following antisymmetry condition [120, 200, 201],

$$F^{\sigma_1,\sigma_2}(\boldsymbol{k}_1, \boldsymbol{k}_2; t_1, t_2) = -F^{\sigma_2,\sigma_1}(\boldsymbol{k}_2, \boldsymbol{k}_1; t_2, t_1). \tag{53}$$

which is the key to the emergence of distinct superconducting symmetries. Since the systems we are interested in are periodic in time, in what follows, we would like to exploit such time-

periodicity within the Floquet framework. In this regard, we expand the anomalous Green's function as [187, 229]

$$F^{\sigma_1,\sigma_2}(\boldsymbol{k}_1,\boldsymbol{k}_2;t_1,t_2) = \sum_{n,m}\int_{-\Omega/2}^{\Omega/2}\frac{d\omega}{2\pi}F^{\sigma_1,\sigma_2}_{n,m}(\boldsymbol{k}_1,\boldsymbol{k}_2;\omega)\,e^{-i(\omega+n\Omega)t_1}\,e^{i(\omega+m\Omega)t_2}, \tag{54}$$

where $\omega \in [-\Omega/2,\Omega/2]$, $\Omega$ is the frequency of the drive, and $F^{\sigma_1,\sigma_2}_{n,m}$ represent the Floquet pair amplitudes between electrons in the $n^{\text{th}}$ and $m^{\text{th}}$ sidebands by the absorption/emission of $|n-m|$ photons. The antisymmetry condition [Eq. (53)], combined with the Floquet expansion [Eq. (54)], imposes the constraint for Floquet pair amplitudes given by

$$F^{\sigma_1,\sigma_2}_{n,m}(\boldsymbol{k}_1,\boldsymbol{k}_2;\omega) = -F^{\sigma_2,\sigma_1}_{-m,-n}(\boldsymbol{k}_2,\boldsymbol{k}_1;-\omega), \tag{55}$$

after a total exchange of Floquet indices $(n,m) \leftrightarrow (-m,-n)$, spins $\sigma_1 \leftrightarrow \sigma_2$, momentum $\boldsymbol{k}_1 \leftrightarrow \boldsymbol{k}_2$, and frequency $\omega \leftrightarrow -\omega$. This generalized antisymmetry condition allows the appearance of eight distinct classes of Floquet Cooper pairs, comprising four spin-singlet and four spin-triplet types, thus broadening the symmetries of superconducting correlations in Floquet systems [187].

   In the BdG formalism employed here [Eq. (8)], we simplify notation by introducing the index $\nu$ to denote the spin configuration of Cooper pairs, rather than explicitly writing $(\sigma_1,\sigma_2)$. Specifically, $\nu = + (-)$ labels spin-up (spin-down) electrons paired with spin-down (spin-up) holes. As noted at the beginning of this subsection, the Floquet pair amplitudes are encoded in the anomalous components of the Floquet Green's function in Nambu space, associated with the Floquet BdG Hamiltonian via [187]. More precisely, the Floquet Nambu Green's function is obtained as

$$\hat{G}^{\nu}_F(\omega,\boldsymbol{k}) = \left[\omega - \mathcal{H}^{\nu}_F(\boldsymbol{k})\right]^{-1} = \begin{pmatrix} G^{\nu}_F(\omega,\boldsymbol{k}) & F^{\nu}(\omega,\boldsymbol{k}) \\ \bar{F}^{\nu}(\omega,\boldsymbol{k}) & \bar{G}^{\nu}_F(\omega,\boldsymbol{k}) \end{pmatrix}, \tag{56}$$

where $G^{\nu}_F$ and $\bar{G}^{\nu}_F$ are the normal (electron-electron and hole-hole) Green's functions, while $F^{\nu}$ and $\bar{F}^{\nu}$ denote the anomalous (electron-hole and hole-electron) components. The Floquet pair amplitudes $F^{\nu}_{n,m}(\omega,\boldsymbol{k})$ correspond to the $(n,m)$ Fourier components of the anomalous block $F^{\nu}(\omega,\boldsymbol{k})$. As noted above, the symmetry of the Floquet pair amplitudes gives the type of the Cooper pairs, which requires identifying symmetry classes fulfilling the antisymmetry condition under the total exchange of the quantum numbers of the paired electrons. Under the exchange of Floquet indices, we find

$$F^{\nu,\pm}_{n,m}(\omega,\boldsymbol{k}) = \frac{1}{2}\left[F^{\nu}_{n,m}(\omega,\boldsymbol{k}) \pm F^{\nu}_{-m,-n}(\omega,\boldsymbol{k})\right], \tag{57}$$

where the transformation $(n,m) \leftrightarrow (-m,-n)$ reflects the exchange symmetry between the two Floquet indices [187]. Below, we refer to $F^{\nu,+}_{n,m}$ and $F^{\nu,-}_{n,m}$ as even-Floquet and odd-Floquet, respectively. To separate spin symmetry, we further classify spin-singlet and spin-triplet pair amplitudes as [218]

$$F^{s,\pm}_{n,m}(\omega,\boldsymbol{k}) = \frac{1}{2}\left[F^{\nu,\pm}_{n,m}(\omega,\boldsymbol{k}) - F^{-\nu,\pm}_{n,m}(\omega,\boldsymbol{k})\right], \tag{58}$$

$$F^{t,\pm}_{n,m}(\omega,\boldsymbol{k}) = \frac{1}{2}\left[F^{\nu,\pm}_{n,m}(\omega,\boldsymbol{k}) + F^{-\nu,\pm}_{n,m}(\omega,\boldsymbol{k})\right], \tag{59}$$

where $F^{s,+}_{n,m}$ ($F^{s,-}_{n,m}$) correspond to spin-singlet even-Floquet (odd-Floquet) components, while $F^{t,+}_{n,m}$ ($F^{t,-}_{n,m}$) denote spin-triplet even-Floquet (odd-Floquet) pairings. Similarly, we can decompose the symmetry in frequency and momentum, where the Floquet pair amplitudes can be

even/odd under the individual exchange of frequency and momentum. Thus, this decomposition enables a systematic analysis of all Cooper pair symmetry classes in time-periodically driven $d$- and $p$-wave unconventional magnets with conventional superconductivity. By analyzing the Floquet pair amplitudes, we identify eight distinct classes of Floquet Cooper pairs that fulfill the antisymmetry condition given by Eq. (55) under the exchange of spins, Floquet indices, frequency, and momentum. These Floquet pair symmetry classes are summarized in Table 1.

More specifically, we define the amplitude for each pair symmetry class (C) by summing the corresponding Floquet components, where pairing occurs between electrons residing in the same Floquet band or between Floquet bands; the processes happen by the emission and absorption of an even/odd number of photons. Taking this into account, for an even number of photons processes, the pair amplitudes are described by $F_{n,n+2m}^{s/t,\pm}$; the $m = 0$ case represents intra-sideband pairing while $m \neq 0$ corresponds to inter-sideband pairing, both involving an even number of photons. Similarly, for odd-number photon processes, the pair amplitudes between Floquet bands are given by $F_{n,n+2m+1}^{s/t,\pm}$, where only inter-sideband pairing is involved. Therefore, there exist two types of Floquet pair amplitudes that become relevant, namely, $F_{n,n+2m}^{s/t,\pm}$ and $F_{n,n+2m+1}^{s/t,\pm}$. While $F_{n,n+2m}^{s/t,\pm}$ represents pairing between electrons in the $n^{\text{th}}$ and $(n+2m)^{\text{th}}$ Floquet sidebands, $F_{n,n+2m+1}^{s/t,\pm}$ characterizes the pairing between $n^{\text{th}}$ and $(n+2m+1)^{\text{th}}$ sidebands. As already mentioned above, these Floquet pairings characterize the emergence of Floquet Cooper pairs via the absorption or emission of $|2m|$ and $|2m+1|$ photons, which correspond to an even and odd number of photons, respectively. Since the Floquet pair amplitudes ($F_{n,n+2m}^{s/t,\pm}$ and $F_{n,n+2m+1}^{s/t,\pm}$) determine the type of Floquet pair symmetry classes, below we write them all for unconventional magnets with conventional superconductors. For $d$- and $p$-wave magnets, we obtain four spin-singlet classes given by

$$F_{C1}^{s,j} = \sum_{n,m} F_{n,n+2m}^{s,+} = F_{0\Omega}^{s,+} + F_{2\Omega}^{s,+} + \cdots,$$

$$F_{C2}^{s,j} = \sum_{n,m} F_{n,n+2m+1}^{s,+} = F_{1\Omega}^{s,+} + F_{3\Omega}^{s,+} + \cdots,$$

$$F_{C3}^{s,j} = \sum_{n,m} F_{n,n+2m}^{s,-} = F_{0\Omega}^{s,-} + F_{2\Omega}^{s,-} + \cdots, \tag{60}$$

$$F_{C4}^{s,j} = \sum_{n,m} F_{n,n+2m+1}^{s,-} = F_{1\Omega}^{s,-} + F_{3\Omega}^{s,-} + \cdots,$$

where $j = \{d, p\}$ and the employed Floquet components are assumed to correspond to each type of unconventional magnet, although for simplicity we do not explicitly write the $j$ index in the Floquet components. Note that the Floquet pair amplitudes after the second equality are written as $F_{|2m|\Omega}^{s,\pm}$ and $F_{|2m+1|\Omega}^{s,\pm}$, where the subscript index labels the difference between the two Floquet indices in $F_{n,n+2m}^{s,\pm}$ and $F_{n,n+2m+1}^{s,\pm}$, respectively. This way helps us understand the even/odd number of photons involved in the emergence of the distinct Floquet pair symmetry classes.

For the $d$-wave altermagnets, we find four spin-triplet classes given by

$$F_{C5}^{t,d} = \sum_{n,m} F_{n,n+2m}^{t,+} = F_{0\Omega}^{t,+} + F_{2\Omega}^{t,+} + \cdots,$$

$$F_{C6}^{t,d} = \sum_{n,m} F_{n,n+2m+1}^{t,+} = F_{1\Omega}^{t,+} + F_{3\Omega}^{t,+} + \cdots,$$

$$F_{C7}^{t,d} = \sum_{n,m} F_{n,n+2m}^{t,-} = F_{0\Omega}^{t,-} + F_{2\Omega}^{t,-} + \cdots, \tag{61}$$

$$F_{C8}^{t,d} = \sum_{n,m} F_{n,n+2m+1}^{t,-} = F_{1\Omega}^{t,-} + F_{3\Omega}^{t,-} + \cdots,$$

Table 1: Pair symmetry classes of the Floquet pair amplitudes in $d$-wave ($p$-wave) unconventional magnets with spin-singlet $s$-wave superconductivity under light drives. Both $d$- and $p$-wave systems host identical classification for the spin-singlet classes (1–4), but the spin-triplet classes (5–8) switch depending on the momentum parity of the underlying unconventional magnetism. The symmetries related to $d$-wave ($p$-wave) unconventional magnets are also valid for higher angular momentum even-parity (odd-parity) unconventional magnets.

| Floquet Components | Spin $\sigma_1 \leftrightarrow \sigma_2$ | Floquet $(n,m) \leftrightarrow (-m,-n)$ | Frequency $\omega \leftrightarrow -\omega$ | Momentum $k_1 \leftrightarrow k_2$ | Class |
|---|---|---|---|---|---|
| $F_{n,n+2m}^{s,+}$ | Singlet | Even | Even | Even | 1 |
| $F_{n,n+2m+1}^{s,+}$ | Singlet | Even | Odd | Odd | 2 |
| $F_{n,n+2m}^{s,-}$ ($m \neq 0$) | Singlet | Odd | Odd | Even | 3 |
| $F_{n,n+2m+1}^{s,-}$ | Singlet | Odd | Even | Odd | 4 |
| $F_{n,n+2m}^{t,+}$ | Triplet | Even | Odd (Even) | Even (Odd) | 5 (6) |
| $F_{n,n+2m+1}^{t,+}$ | Triplet | Even | Even (Odd) | Odd (Even) | 6 (5) |
| $F_{n,n+2m}^{t,-}$ ($m \neq 0$) | Triplet | Odd | Even (Odd) | Even (Odd) | 7 (8) |
| $F_{n,n+2m+1}^{t,-}$ | Triplet | Odd | Odd (Even) | Odd (Even) | 8 (7) |

and also four spin-triplet classes for $p$-wave magnets given by

$$
\begin{aligned}
F_{C5}^{t,p} &= \sum_{n,m} F_{n,n+2m+1}^{t,+} = F_{1\Omega}^{t,+} + F_{3\Omega}^{t,+} + \cdots, \\
F_{C6}^{t,p} &= \sum_{n,m} F_{n,n+2m}^{t,+} = F_{0\Omega}^{t,+} + F_{2\Omega}^{t,+} + \cdots, \\
F_{C7}^{t,p} &= \sum_{n,m} F_{n,n+2m+1}^{t,-} = F_{1\Omega}^{t,-} + F_{3\Omega}^{t,-} + \cdots, \\
F_{C8}^{t,p} &= \sum_{n,m} F_{n,n+2m}^{t,-} = F_{0\Omega}^{t,-} + F_{2\Omega}^{t,-} + \cdots.
\end{aligned}
\tag{62}
$$

Thus, $F_{C1,C3}^{s,j}$, $F_{C5,C7}^{t,d}$, and $F_{C6,C8}^{t,p}$ correspond to Floquet Cooper pair amplitudes involving an even number of photon processes, while $F_{C2,C4}^{s,j}$, $F_{C6,C8}^{t,d}$, and $F_{C5,C7}^{t,p}$ involve an odd number of photon processes. Depending on the momentum parity, the frequency symmetry might differ in $d$- and $p$-wave magnets, see Table 1. Notably, the zero-photon components $F_{0\Omega}^{s(t),\pm}$ remain finite even in static systems when the driving is absent. In contrast, the dominant components for odd-photon processes, such as $F_{1\Omega}^{s(t),\pm}$, require at least one photon to facilitate pairing, triggered by the drive. We therefore conclude that all pair amplitudes involving odd-photon processes are induced by the driving field, while their even-photon counterparts can exist even in the absence of periodic driving. The even/odd number of photons involved in the pair amplitudes also have an effect on the type of spin symmetries they acquire (Table 1). For instance, the spin-singlet pair amplitudes in $p$- and $d$-wave unconventional magnets can be formed by involving an even (or odd) number of photon processes, see Eqs. (60). In contrast, to form spin-triplet pair amplitudes, the involved number of photon processes depends on the type of unconventional magnet, which can lead to Floquet Cooper pairs with different symmetry classes but using the same photon parity, see Eqs. (61) and (62).

An example of the above discussion is that the even-photon processes induce spin-singlet pair symmetries that belong to spin-singlet classes 1 and 3 in both driven $d$- and $p$-wave magnet; but the same even photon parity leads to spin-triplet pairs belonging to classes 5 and 7 in

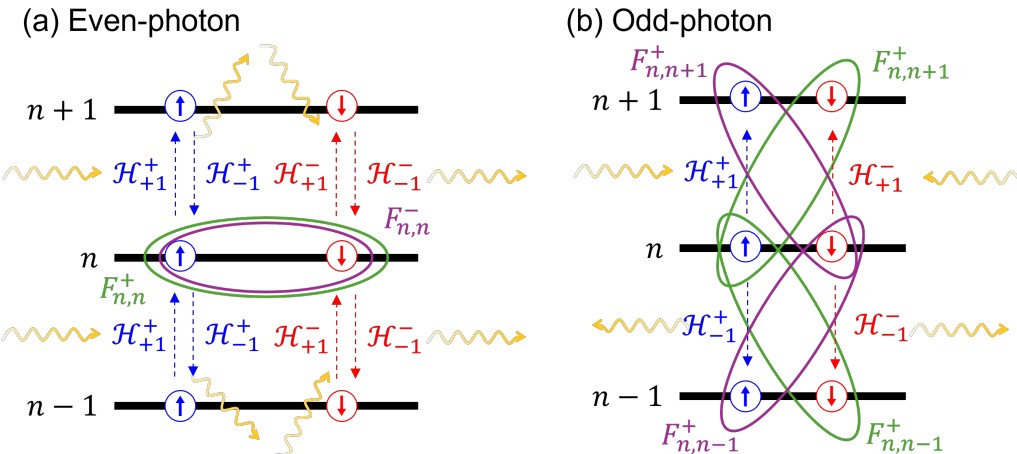

Figure 6: Illustration of Floquet Cooper pair amplitudes arising from photon-assisted processes. (a) Even-photon processes involve virtual transitions between Floquet sidebands that preserve the initial and final sideband indices ($F_{n,n}^{\nu}$), mediated by, e.g., processes of abosorbing one photon transitting from the $n^{\text{th}}$ to $(n+1)^{\text{th}}$ Floquet sideband via $\mathcal{H}_{+1}^{+}$, then emitting one jumping back to the $n^{\text{th}}$ Floquet sideband via $\mathcal{H}_{+1}^{-}$. (b) Odd-photon processes involve virtual transitions between neighboring Floquet sidebands, mediated by single-photon processes $\mathcal{H}_{\pm 1}^{\pm}$, producing off-diagonal pairing amplitudes such as $F_{n,n\pm 1}^{\nu}$. Blue and red circles with arrows indicate spin-up and spin-down electrons, respectively. The thin dashed arrows represent photon absorption or emission, while wavy lines denote the photons. Pair amplitudes $F_{n,m}^{\nu}$ with $\nu = +$ ($\nu = -$) characterize Cooper pairs between electrons with spin up (spin down) and electrons with spin down (spin up) are encircled by green (violet) ellipses.

a driven $d$-wave altermagnet or classes 6 and 8 in a driven $p$-wave magnet, see Table 1. This different classification in spin-triplet Floquet Cooper pairs originates from the contrasting parity of $d$-wave and $p$-wave magnetic orders, which, as already anticipated, relates to the parity of photons involved in forming Cooper pairs. To understand this effect, let us consider the case with $m = 0$ as an example, with a schematic diagram shown in Fig. 6. Under general conditions, intra-sideband and inter-sideband Floquet pair amplitudes involve the contribution of different Floquet bands and therefore involve a different number of photons; these photon-assisted processes can be formally derived from Dyson's perturbation expansion [187], but here we simply use the diagram shown in Fig. 6. For instance, the intra-sideband Floquet pair amplitude ($F_{n,n}^{\nu}$) can appear due to electrons pairing without undergoing transitions to any other sideband, representing a first-order correction that does not involve the emission or absorption of photons; this process leads to the Floquet Cooper pair indicated the greens ellipse in Fig. 6. Since light couples differently to $d$- and $p$-wave unconventional magnets, the momentum parity of such Floquet Cooper pairs is necessarily different, which is why there exists a difference between classes 5 and 7 in a driven $d$-wave altermagnet and classes 6 and 8 in $p$-wave magnets. When nearest-neighbor sidebands come into play, the electron forming the Cooper pairs can transition onto those sidebands and then return back the $n$ sideband thanks to $\mathcal{H}_{\pm 1}^{\nu}$, thereby emitting and absorbing a photon, respectively; see the blue and red dashed arrows as well as wiggle yellow arrows in Fig. 6(a). As a result, the Floquet pair amplitude $F_{n,n}^{\nu}$ gets a contribution from these photon-assisted processes whose number is even; for the next-nearest-neighbor sidebands (and beyond), similar even photon processes occur, but the number increases. As already noted before, the spin-singlet symmetry stems from the interplay between light and the spin-singlet parent superconductor but without any effect due to

unconventional magnets; their existence involving an even number of photons is tied to the finite value of the Floquet components $\mathcal{H}_{\pm 1}^{\gamma}$ and $\mathcal{H}_{\pm 2}^{\gamma}$ in Eqs. (35-36). For this reason, the resulting classification of spin-singlet Floquet pair amplitudes for $d$- and $p$-wave unconventional magnets is identical, see Eqs. (60). For the spin-triplet Floquet pairs, however, unconventional magnetism is necessary, and the Floquet bands ensure their existence due to Floquet transitions involving an even number of photons in this case.

A similar discussion can be done for the formation of Floquet pair amplitude due to an odd number of photon processes, as schematically shown in Fig. 6(b). The finite coupling between Floquet sidebands gives rise to spin-singlet classes 2 and 4 in both driven unconventional magnets [Eqs. (60)], while to spin triplet classes 6 and 8 in driven $d$-wave altermagnets [Eqs. (61)] and spin-triplet classes 5 and 7 in driven $p$-wave magnets [Eqs. (62)]. However, unlike the even-photon processes enabling both intra-sideband and inter-sideband Floquet Cooper pairs, only inter-sideband pairing is allowed in the odd-photon processes. This can be easily understood by noting that the $m = 0$ component of the generic Floquet pair amplitude $F_{n,n+2m+1}^{\gamma}$, namely, $F_{n,n\pm 1}^{\gamma}$, is formed by pairing electrons from $n^{\text{th}}$ and $(n \pm 1)^{\text{th}}$ Floquet sidebands; this process involves the absorption or emission of one photon via $\mathcal{H}_{\pm 1}^{\gamma}$, see the blue and red dashed arrow and as well as wiggle yellow arrows in Fig. 6(b). This represents a first-order correction of Dyson's perturbation expansion [187] that involves the emission or absorption of one single photon, and gives rise to inter-sideband Floquet Cooper pairs indicated by the green and violet ellipses in Fig. 6(b). By including more sidebands, a higher odd number of photons processes participate and contribute to the Floquet pair amplitude $F_{n,n+2m+1}^{\gamma}$: allowing for the sideband $n-1$, the pair amplitude $F_{n,n+1}^{\gamma}$ acquires the contribution of a pair amplitude involving three photons, which result from a transition to the sideband $n-1$ by emitting one photon assisted by $\mathcal{H}_{-1}^{\gamma}$ and then transits to the $(n + 1)^{\text{th}}$ Floquet sideband by absorbing two photons via $\mathcal{H}_{+2}^{\gamma}$. The resulting pairing involves a total of three photons, which leads to the second-order correction of Dyson's perturbation expansion [187]. As a result, the Floquet pair amplitude $F_{n,n\pm 1}^{\gamma}$ gets a contribution from these photon-assisted processes whose number is odd; for the pairing involving higher sidebands, similar odd photon processes occur, but the odd number of photons increases. We remark that the Floquet pairs with odd number of photons with spin-singlet symmetry (classes 2 and 4) result from the interplay between light and the parent superconductor; while this might seem similar to the even photon case of the previous paragraph, the odd-photon Floquet pairs do not exist in the static regime since they require at least one photon to exist. Moreover, the spin-triplet Floquet Cooper pairs due to an odd number of photons entirely result from the effect of light, unconventional magnetism, and conventional superconductivity, but, unlike the even-photon pairs, these odd-Floquet pairs do not exist in the static regime. Their origin comes from the non-trivial light matter interactions, i.e. $\alpha_d k_A k$ in $\mathcal{H}_{\pm 1}^{d,\gamma}$ and $\alpha_p k_A$ in $\mathcal{H}_{\pm 1}^{p,\gamma}$, see Eqs. (35-36), which demonstrate a distinct momentum parity inherited from the unconventional magnetism: odd-parity for the driven $d$-wave magnet but even-parity for the driven $p$-wave magnet. This momentum parity, accompanied by the Floquet indices, frequency, and present spins, results in the symmetry classification of the spin-triplet Floquet Cooper pairs depending on the type of unconventional magnet, even if the same odd number of photons is involved.

Moreover, Fig. 6 illustrates that the Cooper pairs mediated by even-photon processes involve pairing within the same Floquet sideband, which can already exist in the static (non-driven) system (zero-photon); since at least one photon is involved, the odd-photon processes indicate the necessity of the driving field. Fig. 6(a) demonstrates the example of forming Cooper pairs in the $n = 0$ Floquet sideband. For pairing with an electron in the $n = 0$ Floquet sideband, the partner electron may either interact directly without involving photons, corresponding to the static zero-photon process, or undergo a two-photon process in which it absorbs a photon to reach the $n = +1$ sideband and subsequently emits a photon to return

to $n = 0$ before pairing within the $n = 0$ sideband. As a result, even-photon processes naturally mix with equilibrium pairing correlations. In contrast, as shown in Fig. 6(b), odd-photon processes necessarily involve pairing between different Floquet sidebands. The lowest-order odd-photon contribution corresponds to a one-photon process, in which an electron in the $n = 0$ Floquet sideband absorbs a photon and pairs with an electron in the $n = +1$ sideband. Such pairing channels vanish in the absence of the driving field and therefore represent purely light-induced correlations. This distinction explains why odd-photon processes generate pairing channels that have no static counterpart.

Thus, the momentum parities inherited from the underlying unconventional magnet, accompanied by the Floquet indices, frequency, and present spins, determine the type of emergent Floquet Cooper pair symmetry classes listed in Table 1. Based on this classification, we conclude that among the eight symmetry classes of Cooper pairs listed in Table 1, two spin-singlet classes, $F_{C2(C4)}^{s,j}$, are induced solely by the driving field. Two spin-triplet classes, $F_{C6(C8)}^{t,d}$ for the $d$-wave altermagnet and $F_{C5(C7)}^{t,p}$ for the $p$-wave magnet, arise from the interplay between the drive and the underlying unconventional magnetism. The remaining triplet and singlet pair symmetry classes for each unconventional magnet are pair correlations that originate from those in the static regime, see Subsec. 2.2. In the following, we analyze several specific Floquet Cooper pair amplitudes to characterize (i) Cooper pairs induced by the drive [Sec. 5.2.1], (ii) identify the role of multi-photon processes [Sec. 5.2.2], and (iii) unveil the impact of linearly polarized light.

### 5.2.1 Inspecting the symmetries of Floquet Cooper pairs

As already anticipated above, the eight Floquet Cooper pair symmetries presented in Table 1 exhibit a certain symmetry with respect to the exchange of spins, Floquet indices, frequency, and momentum. Out of these classes, the interplay of drive and unconventional magnetism promotes two classes of spin-singlet Cooper pairs possessing odd parity, which correspond to classes 2 and 4 in Table 1 and emerge due to an odd number of photon processes. These two classes are odd (even) functions under the exchange of Floquet indices, which is tied to the evenness (oddness) under frequency in order to fulfill the antisymmetry condition. The odd-frequency and odd-parity symmetries of $F_{C2}^{s,d}$ are demonstrated in Fig. 7(a,b) for $d_{x^2-y^2}$-wave altermagnet with $\theta_J = 0$ under CPL, see arrows in Fig. 7(a,b). The pair symmetry class $F_{C2}^{s,d}$, with odd-frequency spin-singlet odd-parity, is unexpected in the static case without breaking translational invariance [200], but here it appears thanks to the additional quantum number offered by the Floquet sidebands [Eq. (53)]. The same behavior is observed in the $d_{xy}$-wave magnet with $\theta_J = \pi/4$ under CPL, as shown in Fig. 7(c,d). We have also verified the corresponding symmetries of the pair symmetry class $F_{C4}^{s,d}$, which exhibits a spin-singlet odd-Floquet even-frequency odd-parity symmetry and arises due to an odd number of photon processes like $F_{C2}^{s,d}$. As for the Floquet pair classes 2 and 4 in $p$-wave magnets, they possess the same symmetries as for $d$-wave altermagnets and their dependences are demonstrated in Figs. 8(a,b) for $p_x$- and $p_y$-wave magnets.

In addition to the spin-singlet Cooper pairs, Floquet unconventional magnets also host four spin-triplet pair symmetry classes, with two classes entirely coming from the interplay among light, unconventional magnetism, and conventional superconductivity. These emerging spin-triplet Cooper pairs are characterized by classes 6 and 8 in Table 1 and require an odd number of photons. These Floquet spin-triplet classes are odd in momentum and develop even-frequency (class 6) or odd-frequency (class 8) symmetries depending on the symmetry under exchange of Floquet indices. Similar to spin-triplet Cooper pairs in the static state, these Floquet spin-triplet pairs originate from the $d$-wave magnetism [6,72,74]. However, unlike the static case, the Floquet spin-triplet pairs additionally require odd-photon processes [Eq. (61)],

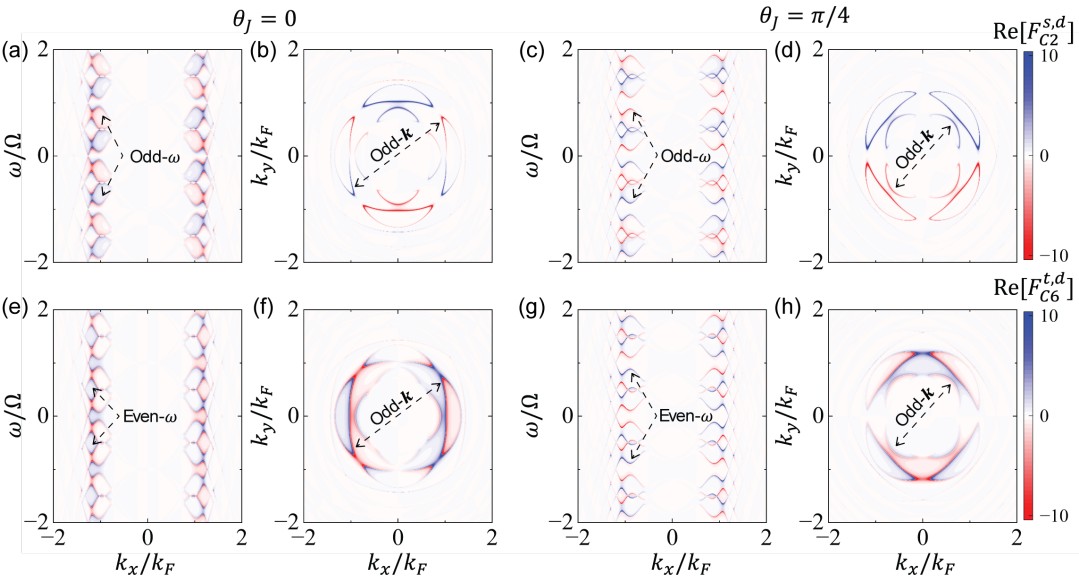

Figure 7: (a,c,e,g) Real part of Floquet Cooper pair amplitudes ($F_{C2}^{s,d}$ and $F_{C6}^{t,d}$) as a function of $\omega$ and $k_x$ in $d_{x^2-y^2}$-wave altermagnets with $\theta_J = 0$ (a,c) and $d_{xy}$-wave magnets with $\theta_J = \pi/4$ (e,g), both under CPL. In (a,e), $k_y = 0$ is fixed, while in (c,g) $k_y/k_F = 0.5$. (b,d,f,h) The same quantity as in (a,c,e,g) but as a function of $k_x$ and $k_y$ at fixed frequency $\omega = 0.5\Omega$. Black dashed arrows in all panels indicate the evenness and oddness of the pair amplitudes with respect to frequency and momentum. Other parameters: $B = 1$, $k_F = 1$, $\mu = 1$, $k_A/k_F = 0.5$, $\alpha_d = 0.2$, and $\Delta = 1.5\Delta_c$.

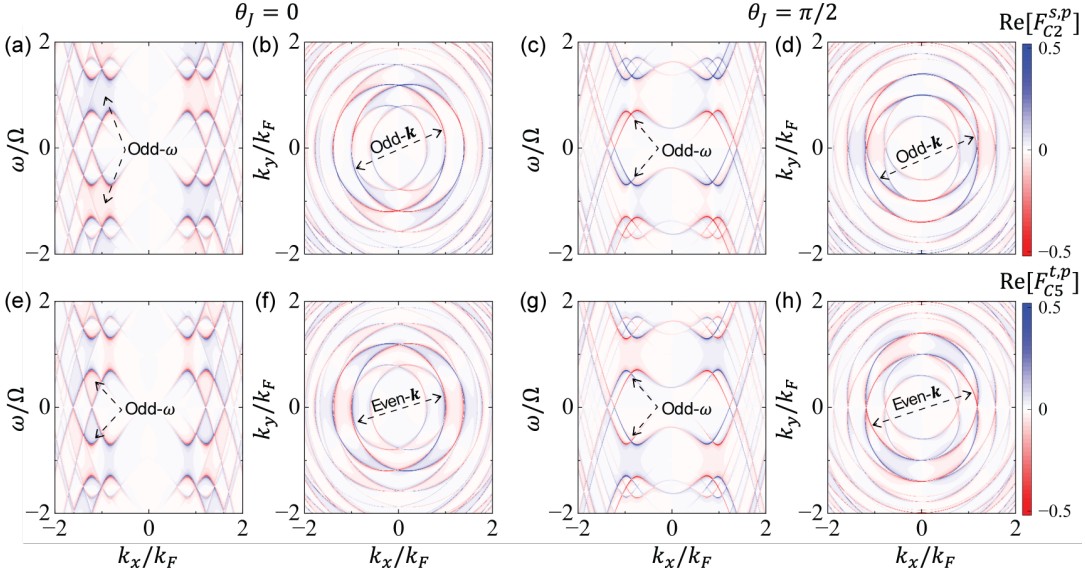

Figure 8: (a,c,e,g) Real part of the Floquet Cooper pair amplitudes ($F_{C2}^{s,p}$ and $F_{C5}^{t,p}$) as functions of $\omega$ and $k_x$ in $p_x$-wave magnets with $\theta_J = 0$ (a,c) and $p_y$-wave magnets with $\theta_J = \pi/2$ (e,g), both under CPL. In (a,e), $k_y = 0$, while in (c,g) $k_y/k_F = 0.5$. (b,d,f,h) The same quantity as in (a,c,e,g) but as a function of $k_x$ and $k_y$ at fixed frequency $\omega = 0.5\Omega$. Black dashed arrows in all panels indicate the evenness and oddness of the pair amplitudes with respect to frequency and momentum. Other parameters: $B = 1$, $k_F = 1$, $\mu = 1$, $k_A/k_F = 0.5$, $\alpha_p = 0.2$, and $\Delta = 0.3\mu$.

making their formation entirely due to the interplay among light, $d$-wave altermagnetism, and conventional superconductivity. Figs. 7(e-h) show the frequency and momentum dependences of $F_{C6}^{t,d}$ for $d_{x^2-y^2}$- and $d_{x,y}$ altermagnets under CPL, which are consistent with Table 1. In the case of $p$-wave magnets shown in Fig. 8, we find that the classes 5 and 7 entirely appear due to the effect of light on $p$-wave magnets with conventional superconductivity: these classes exhibit spin-singlet even-Floquet (odd-Floquet), even-frequency (odd-frequency), and even-parity symmetry. The evenness in parity of the Floquet spin-triplet pairs in $p$-wave magnets is opposite to what occurs in driven $d$-wave altermagnets and also different from the static spin-triplet pairs in $p$-wave magnets, which inherently carry odd parity due to the odd parity of $p$-wave magnets. Under the light drive, however, the evenness in momentum is compensated by the symmetry due to the Floquet indices, which also adjusts the frequency dependence in order to fulfill Fermi-Dirac statistics.

As a result, the spin-triplet Floquet Cooper pair symmetries (classes 6 and 8 for $d$-wave and 5 and 7 for $p$-wave) are intrinsic to the interplay between the time-periodic drive and the unconventional magnetism with spin-singlet $s$-wave superconductivity. Overall, this expanded classification demonstrates how periodic driving can unlock novel superconducting pairing channels, offering new avenues to manipulate and engineer superconducting states in unconventional magnetic systems.

### 5.2.2 Photon contribution

We have seen that, depending on the number of photon processes involved in the formation of Floquet pair amplitudes, the emergent Cooper pairs acquire distinct symmetries. This is better seen in the pair amplitudes written as $F_{N\Omega}^{s(t),\pm}$ after the second equality in Eqs. (60)-(62), where $N$ here labels the number of photons involved in the emergence of Floquet Cooper pairs. By looking at the Table 1 for $d$-wave altermagnets, we clearly identify that classes 1, 3, 5, and 7 involve an even number of photons since $N = |2m|$; an odd number of photons are involved for classes 2, 4, 6, and 8 since $N = |2m+1|$. In the case of $p$-wave magnets, classes 1, 3, 6, and 8 involve an even number of photons, while classes 2, 4, 5, and 7 require an odd number of photons, see Table 1. To quantify the contributions of the respective Floquet pair amplitudes with a certain number of photons, we compare the magnitude of the respective classes with the contribution coming from the lowest photon process. We focus on two representative classes in $d$- and $p$-wave unconventional magnets: $|F_{C1}^{s,d}|$ and $|F_{C6}^{t,d}|$ in $d$-wave altermagnets, while on $|F_{C1}^{s,p}|$ and $|F_{C5}^{t,p}|$ in $p$-wave magnets.

In Fig. 9(a,b), we show $|F_{C1}^{s,d}|$ and its zero-photon component $|F_{0\Omega}^{s,+}|$ as functions of frequency $\omega$ and momentum $k_x$ at $k_y = 0$ in a $d_{x^2-y^2}$-wave altermagnet under CPL. Both $|F_{C1}^{s,d}|$ and $|F_{0\Omega}^{s,+}|$ display Floquet replicas due to Floquet sidebands, which imposes a periodicity in frequency space such that $|F_{C1}(\omega)| = |F_{C1}(\omega + n\Omega)|$, see also Fig. 7. Both quantities reveal multiple superconducting gaps arising from coupling between different Floquet sidebands and exhibit nearly the same intensity, which further suggests the dominant contribution of the zero-photon pair amplitude ($F_{0\Omega}^{s,+}$) to the total Floquet pair class ($F_{C1}^{s,d}$). This is further supported by noting that the largest of these induced gaps appears near $\omega = n\Omega$, where the zero-photon contribution dominates. This dominant behavior of $F_{0\Omega}^{s,+}$ arises because it is determined by $\mathcal{H}_0^\nu$ [Eq. (35) and Eq. (36)], which is related to the static cases, and hence insentive to the driving, while pairings with higher even number of photons, such as for $|F_{2\Omega}^{s,+}|$, scale with higher even powers in $k_A$ which then give a tiny contribution at reasonable frequency of the drive $\Omega$. Further insights on the role of the drive and $d$-wave altermagnetic strength ($\alpha_d$) can be obtained in Fig. 9(c), where we plot the pair amplitude class 1 integrated in momentum $|\bar{F}_{0\Omega}^{s,+}(\omega)| = \int d\mathbf{k} |F_{0\Omega}^{s,+}(\omega)|/(2\pi)^2$ as a function of $\omega$ and distinct values of $\alpha_d$. Here, we can see that the integrated pair amplitude $|\bar{F}_{0\Omega}^{s,+}|$ is finite without both drive and altermagnetism, but a

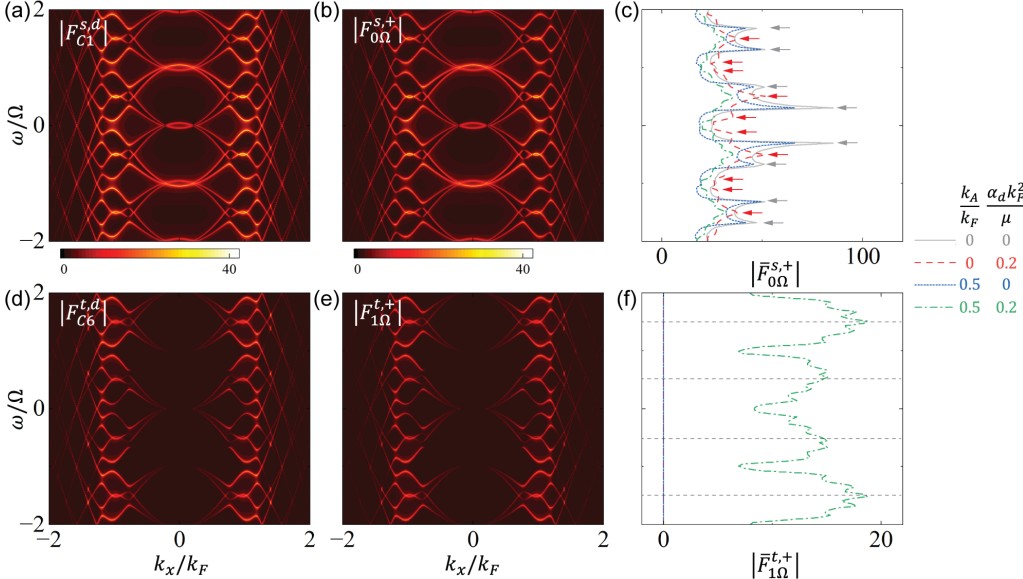

Figure 9: (a,b) Magnitude of the Floquet pair symmetry class $|F_{C1}^{s,d}|$ and its zero-photon component $F_{0\Omega}^{s,+}$ as functions of $\omega$ and $k_x$ for a $d_{x^2-y^2}$-wave altermagnet under CPL at $k_y = 0$. (c) Frequency dependence of the integrated zero-photon component $\bar{F}_{0\Omega}^{s,+}$ for different values of the drive amplitudes $k_A$ and strengths of the $d$-wave altermagnetic field $\alpha_d$. The arrows in (c) indicate peaks in $\bar{F}_{0\Omega}^{s,+}$ corresponding to the superconducting gap edges centered around $\omega = n\Omega$. (d,e,f) The same as in (a,b,c) but for the Floquet pair symmetry class $|F_{C6}^{t,d}|$ and its one-photon component $F_{1\Omega}^{t,+}$. In (f), a vanishing value of either $k_A$ and $\alpha_d$ leads to $\bar{F}_{1\Omega}^{t,+} = 0$, indicated by the vertical line. The horizontal dashed lines indicate that $\bar{F}_{1\Omega}^{t,+}$ is dominant around $\omega = n\Omega/2$, where pairing between neighboring Floquet sidebands occurs. Parameters: same as in Fig. 7.

nonzero value of them introduces interesting features [Fig. 9(c)]. For instance, $|\bar{F}_{0\Omega}^{s,+}|$ exhibits larger values within the first Floquet zone $[-\Omega/2, \Omega/2]$ and develops symmetric peaks below and above $\omega = n\Omega$. Its existence without $d$-wave altermagnetism, depicted by gray and blue curves in Fig. 9(c), is because $|\bar{F}_{0\Omega}^{s,+}|$ has spin-singlet even-parity symmetry (class 1), which is the same symmetry of the parent superconductor. Moreover, the nonzero $|\bar{F}_{0\Omega}^{s,+}|$ without external driving ($k_A = 0$) also explains the dominant contribution of the zero-photon pair amplitude, see gray and red curves in Fig. 9(c). A similar analysis holds for other classes involving an even number of photons. For instance, Fig. 10(a-c) displays the spin-singlet class 1 $|F_{C1}^{s,p}|$ in a $p_x$-wave magnet under a CPL and shows the dominant contribution of its zero-photon component as well as the appearance of induced gaps between Floquet sidebands.

In relation to the Floquet pair amplitudes due to an odd number of photons, in Figs. 9(d,e) we show class 6 $|F_{C6}^{t,d}|$ and its one-photon component $|F_{1\Omega}^{t,+}|$ as functions of $\omega$ and $k_x$ at $k_y = 0$. Unlike the even-photon case in Fig. 9(a), which is distributed across all $k_x$, the odd-photon Floquet pair magnitudes $|F_{C6}^{t,d}|$ and $|F_{1\Omega}^{t,+}|$ develop large intensities around the Fermi points $k_x/k_F \approx 1$, following a very similar behavior that unveils the dominant contribution of $|F_{1\Omega}^{t,+}|$, see Fig. 9(d,e). Integrating this component over momentum yields the quantity $|\bar{F}_{1\Omega}^{t,+}|$, shown in Fig. 9(f) as a function of $\omega$. This integrated pair magnitude reaches maxima near $\omega = n\Omega/2$, where coupling between the $n^{\text{th}}$ and the $(n \pm 1)^{\text{th}}$ Floquet sidebands becomes significant, as indicated by the horizontal dashed lines in Fig. 9(f). We have also verified that $|F_{1\Omega}^{t,+}|$, and the overall class 6 pair magnitude $|F_{C6}^{t,d}|$ arise from the interplay between the driving field ampli-

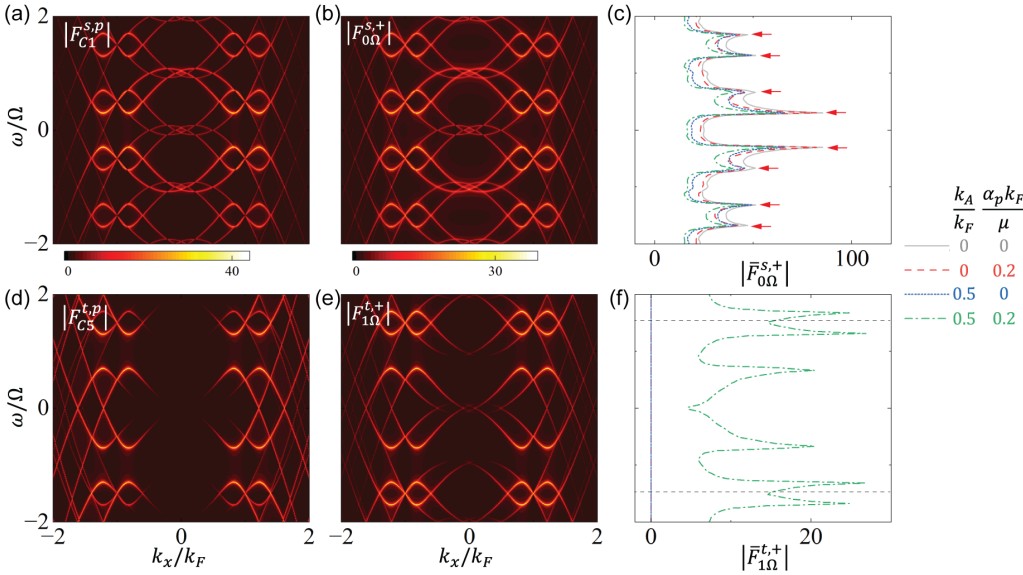

Figure 10: (a,b) Magnitude of the Floquet pair symmetry class $|F_{C1}^{s,p}|$ and its zero-photon component $F_{0\Omega}^{s,+}$ as functions of $\omega$ and $k_x$ for a $p_x$-wave magnet under CPL at $k_y = 0$. (c) Frequency dependence of the integrated zero-photon component $\bar{F}_{0\Omega}^{s,+}$ for different values of the drive amplitudes $k_A$ and strengths of the $d$-wave altermagnetic field $\alpha_p$. The arrows in (c) indicate peaks in $\bar{F}_{0\Omega}^{s,+}$ corresponding to the superconducting gap edges centered around $\omega = n\Omega$. (d,e,f) The same as in (a,b,c) but for the Floquet pair symmetry class $|F_{C5}^{t,p}|$ and its one-photon component $F_{1\Omega}^{t,+}$. In (f), a vanishing value of either $k_A$ and $\alpha_p$ leads to $\bar{F}_{1\Omega}^{t,+} = 0$, indicated by the vertical line. The horizontal dashed lines indicate that $\bar{F}_{1\Omega}^{t,+}$ is dominant around $\omega = n\Omega/2$, where pairing between neighboring Floquet sidebands occurs. Parameters are the same as in Fig. 7. Parameters: same as in Fig. 8.

tude $k_A$ and the strength of $d$-wave magnetism $\alpha_d$, becoming finite only when both parameters are nonzero. This is consistent with our argument that spin-triplet, odd-photon-assisted Cooper pairs require the presence of both the external drive and $d$-wave magnetic order. In the case of $p$-wave magnets, Fig. 10(d-f) shows class 5 $|F_{C5}^{t,p}|$ and its dominant contribution from the lowest one-photon pair amplitude $|F_{1\Omega}^{t,+}|$, which only exist due to the interplay between light and $p$-wave magnetism. A small superconducting gap opens at $\omega = n\Omega/2$, originating from coupling between electrons and holes mediated by the absorption or emission of a single photon, which leads to strong pair amplitudes; see Fig. 10(d,e) and horizontal dashed line in Fig. 10(f).

Overall, these results highlight how, depending on the number of photons, emergent Floquet Cooper pairs in driven unconventional magnets develop distinct dependences. Even-photon processes, such as those contributing to $|F_{C1}|$, largely preserve the properties of the static spin-singlet superconducting state and are relatively insensitive to the drive and magnetism. In contrast, odd-photon-assisted spin-triplet pairings, such as $\left|F_{C5}^{p}\right|$, emerge due to the interplay between the periodic drive and the unconventional magnetic order. These odd-photon contributions manifest distinct features in both frequency and momentum space, and their appearance near $\omega = n\Omega/2$ underscores the critical role of Floquet sideband mixing in enabling new pairing channels. Thus, the number of photon fundamentally shapes the symmetry and spectral characteristics of Cooper pairs in driven unconventional magnets.

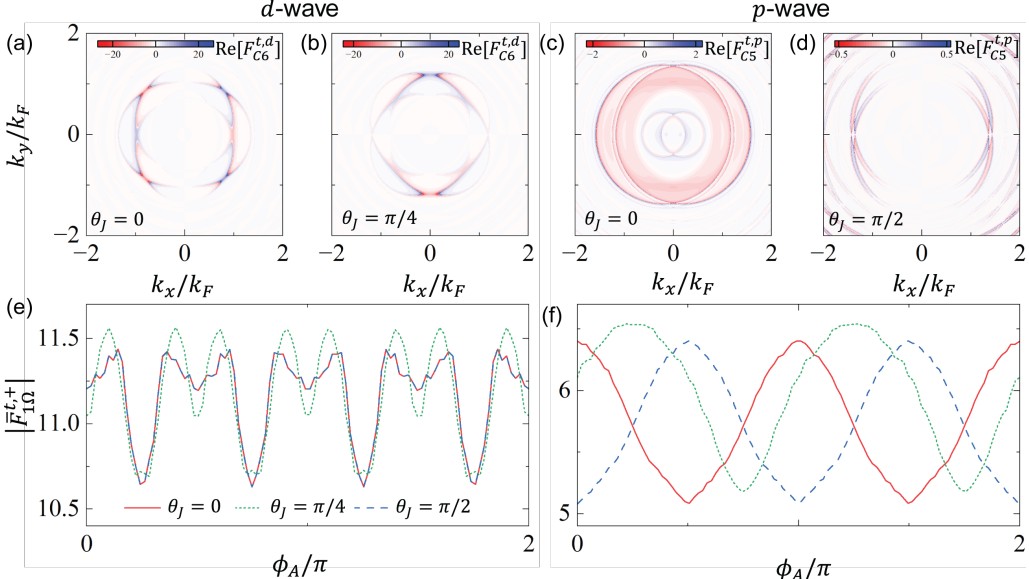

Figure 11: Upper panel: (a) and (b): The light-induced spin-triplet, even-Floquet, even-frequency, odd-momentum Cooper pair amplitude $\mathrm{Re}[F_{C6}^d]$ in $d_{x^2-y^2}$-wave magnets with $\theta_J = 0$ and $d_{xy}$-wave magnets with $\theta_J = \pi/4$, respectively, driven by linearly polarized light with $\phi_A = 0$. (c) and (d): The light-induced spin-triplet, even-Floquet, odd-frequency, even-momentum Cooper pair amplitude $\mathrm{Re}[F_{C5}^p]$ in $p_x$-wave magnets with $\theta_J = 0$ and $p_y$-wave magnets with $\theta_J = \pi/2$, respectively, driven by linearly polarized light with $\phi_A = 0$. In all cases, the momentum parity is preserved. Lower panel: The integrated one-photon contribution to the Cooper pair magnitude, $|\bar{F}_{1\Omega}^{t,+}|$, in driven (e) $d$-wave and (f) $p$-wave magnets, shown for various values of $\theta_J$. Parameters are the same as in Figs. 7 and 8 for the driven $d$-wave and $p$-wave cases, respectively.

### 5.2.3 Effects of linearly polarized light

While the Floquet pair symmetries presented in Table 1 are present in unconventional magnets under CPL and LPL, we have so far explored them under CPL. For this reason, here we would like to inspect further the effect of LPL. We first start by noting that, under LPL, the elements of the Floquet Hamiltonian in Eqs. (35) and (36) depend explicitly on the polarization direction of the LPL field $\phi_A$. To be more precise, in the zero-photon sector of the driven $d$-wave altermagnet with superconductivity ($\mathcal{H}_0^{d,\nu}$), an effective Zeeman field is induced and given by $k_A^2 [\nu \alpha_d \cos(2\theta_J - 2\phi_A)\tau_0]/2$ [Eqs. (35)], which gives rise to a finite spin density and a finite spin-triplet Cooper pair amplitude in the high-frequency limit [75]. Moreover, the components involving single photons are $\mathcal{H}_{+1}^{d,\nu} = \nu k_A k \, \alpha_d \cos(\theta_k - 2\theta_J + \phi_A)\tau_z$, which reflects that they depend sensitively on the relative orientation between the $d$-wave order parameter ($\theta_J$) and the LPL polarization direction ($\phi_A$), see Eqs. (35); these contributions can vanish along specific momentum directions where $\theta_k$ satisfies $\theta_k - 2\theta_J + \phi_A = (2n+1)\pi/2$. Furthermore, the two-photon terms are momentum-independent but also modulated by $\theta_J$ and the LPL polarization angle $\phi_A$ via $\nu k_A^2 \alpha_d \cos(2\theta_J - 2\phi_A)$, see Eqs. (35). Unlike all these $d$-wave Floquet components, in $p$-wave magnets under LPL, the nontrivial light-matter interaction arises primarily from single-photon processes given by $\nu k_A \alpha_p \cos(\theta_J - \phi_A)$ and is momentum-independent, see Eq. (36). These anisotropic couplings between Floquet sidebands due to the polarization angle $\phi_A$ are absent in CPL-driven systems, as discussed in Secs. 5.2.1–5.2.2; see also Figs. 7 and 8.

To visualize a representative Floquet pair amplitude, in Fig. 11(a,b) we show $\mathrm{Re}F^{t,d}_{C6}$ as a function of momenta for $d_{x^2-y^2}$- and $d_{xy}$-wave altermagnets under LPL at $\phi_A = 0$. Here, $\mathrm{Re}F^d_{C6}$ vanishes whenever the condition $\theta_k - 2\theta_J + \phi_A = (2n+1)\pi/2$ is satisfied, indicating the role of LPL and distinct from what is shown in Fig. 7 for $d$-wave altermagnets under CPL. Consequently, in a $d_{x^2-y^2}$-wave magnet with $\theta_J = 0$ driven by linearly polarized light with polarization angle $\phi_A = 0$ (i.e., polarization along the $x$-axis), the Cooper pair amplitude vanishes at $k_x = 0$ (corresponding to $\theta_k = \pi/2$). In contrast, for a $d_{xy}$-wave magnet with $\theta_J = \pi/4$, the Cooper pair amplitude vanishes at $k_y = 0$ (i.e., $\theta_k = 0$). The competition between the LPL polarization direction ($\phi_A$) and the orientation of the unconventional magnetic order ($\theta_J$) is also evident in $p$-wave magnets under LPL, as demonstrated in Figs. 11(c,d), where we plot the momentum dependence of $\mathrm{Re}F^p_{C5}$ at $\theta_J = 0$ and $\theta_J = \pi/2$ for to $p_x$- and $p_y$-wave magnets, respectively. In this case, when the polarization direction is parallel to the $p$-wave magnetic orientation, the effect of the drive becomes pronounced but it is minimal when $\phi_A$ and $\theta_J$ are perpendicular to each other, leading to a vanishing Cooper pair amplitude $\mathrm{Re}F^p_{C5}$ along the polarization direction. This relationship between LPL polarization direction and $p$-wave magnetic orientation is embodied in the nontrivial light-matter coupling in Eq. (36), which takes the form $\nu k_A \alpha_p \cos(\theta_J - \phi_A)$, yielding vanishing coupling when $\theta_J - \phi_A = (2n+1)\pi/2$. Thus, the interplay between magnetic anisotropy and the polarization of the LPL leads to characteristic angular patterns in the induced Cooper pair amplitudes.

Further insights on the interplay between the LPL polarization direction ($\phi_A$) and $\theta_J$ can be obtained from Figs. 11(e,f), where the classes discussed above but integrated in momentum and denoted as $|\bar{F}^{t,d}_{C6}|$ and $|\bar{F}^{t,p}_{C5}|$ for $d$- and $p$-wave unconventional magnets at distinct $\theta_J$. We see that $|\bar{F}^{t,d}_{C6}|$ is largely insensitive to variations of $\theta_J$ and $\phi_A$, which is a result of the integration since it averages over momenta; see Fig. 11(e) and its $y$-axis. It is thus challenging to identify the type of $d$-wave altermagnetism by measuring the response of $|\bar{F}^{t,d}_{C6}|$ with respect to the LPL polarization angle $\phi_A$ in low-frequency LPL drives. In the case of $p$-wave magnets shown in Fig. 11(f), the integrated pair amplitude $|\bar{F}^{t,p}_{C5}|$ acquires a more pronounced dependence with respect to $\phi_A$ and $\theta_J$. The momentum-independence of the single-photon processes in the $p$-wave magnets under LPL [Eq. (36)] enables the orientation $\theta_J$ to introduce a phase shift in $|\bar{F}^p_{C5}|$ as a function of $\phi_A$, which follows the relation $\cos(\theta_J - \phi_A)$; this also allows to identify the orientation angle $\theta_J$ for low frequency LPL drives. We can thus close this part by stressing that the response of Floquet-induced Cooper pairs to LPL reveals rich symmetry properties and provides a way to probe the orientation of unconventional magnetism in superconducting systems.

# 6 Discussion

Sections above have systematically demonstrated the theoretical proposal of Floquet engineering spin density in the normal state and Cooper pair correlation in the superconducting states of unconventional magnets. Below, the experimental accessibility of emergent effects and limitations for the Floquet formalism are discussed in Sec. 6.1 and 6.2, respectively.

## 6.1 Experimental observability and feasibility

The main experimentally accessible results proposed in this work are (i) the Floquet-engineered spin density in the normal state (Figs. 3 and 4) and (ii) the emergent spin-triplet pairing amplitude in the superconducting state (Figs. 9–11). Both effects can be generated by applying a mid-infrared (MIR) pump field [130, 142, 230, 231] to unconventional magnets, where superconductivity can be induced by proximity to a conventional spin-singlet $s$-wave. More-

over, these light-induced quantities can be detected using time- and angle-resolved photoe-mission spectroscopy (TrARPES) [142, 230, 231], and also by time-resolved transport mea-surements [130]. Below, we elaborate on the relevant experimental conditions for detecting the predicted signatures.

### 6.1.1 Generation of Floquet sidebands

The observability of the Floquet-engineered spin density and spin-triplet pairing is controlled by the effectiveness of the light-matter interaction. This interaction is characterized by the dimensionless parameter $k_A/k_F = eE_0/(\hbar\Omega k_F)$, where $e$ is the elementary charge, $\hbar$ is the Planck constant, $\Omega$ is the driving frequency, $E_0$ is the electric-field amplitude, and $k_F$ is the Fermi wave vector, see Eq. (20). The field amplitude is related to the driving intensity via $I = c\epsilon E_0^2/2$, where $c$ is the speed of light and $\epsilon$ is the vacuum permittivity [223]. Hence, suitable choices of $\Omega$, $E_0$, and $I$ are required to generate measurable spin splitting and spin-triplet correlations.

Experimentally, Floquet sidebands are commonly realized using MIR pump pulses [130, 142, 230, 231]. Typical photon energies are $\hbar\Omega \sim 10^2$ meV, corresponding to $\Omega \sim 10$ THz, a driving period $T = 2\pi/\Omega \sim 10^2$ fs, and a wavelength of order $10\,\mu$m. For typical Fermi wave vectors $k_F \sim 10^8$–$10^9\,\text{m}^{-1}$ [232], achieving $k_A/k_F = 0.5$, the value used in the currnet work, requires an electric field $E_0 \sim 10^6$–$10^8$ V/m, corresponding to intensities $I \sim 10^{10}$–$10^{14}$ W/m$^2$. Moreover, throughout the this work, we use $\alpha_{d,p} = 0.5/k_F^2$, corresponding to an altermagnetic strength scale of order 100 meV. Since altermagnetic splittings can reach values up to $\sim 1$ eV [16,17], even lower driving field intensities and frequencies may suffice in realistic materials for measurable signals.

Our calculations assume a spatially homogeneous driving field. This approximation is justified because the typical lateral size of fabricated samples is of order $10^2\,\mu\text{m}^2$ [34, 35], while the pump spot size is typically of comparable or larger area: $10^2\,\mu\text{m} \times 10^2\,\mu\text{m}$ with full width at half maximum, [130,230,233]. The spatially homogeneous assumption is reasonable, and the edge effects can therefore be neglected.

Therefore, the generation of the Floquet sidebands reported in this work is expected to be achieved under realistic conditions. Once the conditions for generating Floquet sidebands are achieved, they immediately imply the emergence of Floquet superconducting pair amplitudes.

### 6.1.2 Detection of Floquet spin density and Cooper pair correlations

The Floquet-engineered spin density in the normal state (Figs. 3 and 4) can be probed using TrARPES with an ultraviolet probe [142,230,231]. TrARPES provides energy- and momentum-resolved spectral information, allowing direct observation of multiple Floquet-induced Fermi surfaces in Figs. 3 and 4. To further resolve the spin splitting explicitly, spin-resolved TrARPES measurements are required [234, 235] to observe the Floquet-engineered spin density.

While TrARPES probes the normal component of the Green's function theoretically, it can-not directly probe the Floquet spin-triplet pairs residing in the anomalous Green's function part [see, e.g., Eq. (9)]. Thus, TrARPES is not an ideal technique to detect the Floquet-induced spin-triplet Cooper pairs. Since the anomalous Green's function is related to the Andreev con-ductance [172, 215, 236–238], we expect that the emergent Floquet spin-triplet pairing can be detected via time-resolved transport experiments [130].

The relation between Andreev conductance and pair amplitude is direct since they satisfy $G_A \propto |F|^2$, where $F$ is the anomalous Green's function containing both even- and odd-photon processes, i.e., $F = F^{\text{odd}-\Omega} + F^{\text{even}-\Omega}$, as illustrated in Fig. 6. The even-photon contribution per-sists in the absence of driving due to the existence of the zero-photon (static-state) component,

whereas the odd-photon contribution is induced by the pump field. Consequently, the photon-induced pairing strength can be estimated from the difference between the conductance in the driven and static states: $|F^{odd-\Omega}|^2 \sim G_A(k_A) - G_A(k_A \to 0) \sim |F^{odd-\Omega} + F^{even-\Omega}|^2 - |F^{even-\Omega}|^2$. which includes both spin-singlet and spin-triplet Cooper pair correlations.

Further inspection of the Andreev conductance can distinguish the contribution from the spin-singlet and triplet states. As shown in Figs. 9 and 10, spin-singlet correlations manifest as symmetric conductance peaks near $eV = n\Omega$, reflecting even-parity pairing, whereas spin-triplet correlations give rise to nearly plateau-like features near $eV = n\Omega/2$, where coupling between adjacent Floquet sidebands becomes significant. Thus, the spin-singlet and triplet correlations contribute to the Andreev conductance in different energy windows, hence offering a solid way for their detection.

Detecting the conductance anisotropy is an alternative way to distinguish the contribution from the spin-singlet and triplet states in the transport measurements [130]. The spin-triplet Cooper pair amplitude is strongly anisotropic due to the directional spin splitting of the unconventional magnet, whereas the spin-singlet component is weakly anisotropic as it scales with the proximity-induced gap $\Delta$, see Eq. (11). These distinct features allow singlet and triplet contributions to be distinguished by comparing conductance measurements along different bias and probe directions.

In summary, the proposed Floquet-induced spin splitting and spin-triplet Cooper pairing can be realized using MIR pump fields with frequencies $\Omega \sim 10\,$THz and intensities $I \sim 10^{10}$–$10^{14}\,$W/m$^2$, which are well within current experimental capabilities [130, 142, 230, 231]. TrARPES provides direct access to the spin-resolved Floquet band structure in the normal state, while time-resolved transport measurements enable detection of the spin-triplet pairing correlations in the superconducting state. TrARPES has already been successfully applied to Floquet-Bloch states in graphene [142, 231], topological surface states [230], and antiferromagnets [239, 240]; Time-resolved transport experiments have been employed to probe Floquet Majorana modes [241] and anomalous Hall conductance [130]. These advances demonstrate that the proposed Floquet spin-triplet Cooper pairs are experimentally accessible under present conditions.

## 6.2 Limitations of the Floquet formalism and mitigation strategies

Finally, we remark that heating and the resulting Floquet band occupation, which indeed constitute a central limitation of the present Floquet formalism. These challenges are intrinsic to Floquet engineering of quantum phases and can be mitigated by extending the duration of the long-lived prethermal states that emerge before the irradiated system is completely heated up and can be described effectively by the Floquet formalism [242]. Below, we clarify the role of prethermalization [126, 242] and discuss how heating effects and the redistribution of Floquet states can be minimized in realistic experimental settings [243, 244].

In a closed periodically driven system, photon absorption leads to heating of the electronic systems. After an initial relaxation stage, the system may approach an infinite-temperature-like quasisteady state, characterized by maximal local entropy density subject to conservation laws, such that electrons in the driving system are distributed in all Floquet sidebands. In this regime, nonuniversal details of the Floquet spectrum within each sideband, such as the spin density and pairing, are washed out, while global features remain observable [126].

To avoid heating into the infinite-temperature regime while retaining experimentally observable Floquet engineering effects with minimized electron redistribution, the temporal profile of the driving pulse must be carefully designed to achieve the prethermalized regime [243, 244]: (i) The pulse must be sufficiently short to generate the desired Floquet spin density and spin-triplet correlations before significant heating sets in; (ii) It must be, however, long enough for the system to enter a regime that can be described by a Hamiltonian in the so-called

prethermalized regime. Furthermore, to approximate the ground state of the prethermalized Hamiltonian, the driving field should be switched on adiabatically, favoring a slowly rising pulse with an appropriate duration. Although a complete description of the prethermalized Hamiltonian remains challenging [126], the Floquet formalism used in this work provides a reliable, effective description of the prethermalized state [242], consistent with experimental observations [130, 142, 230, 231].

In typical TrARPES experiments employing a mid-infrared (MIR) pump ($T \sim 100\,\text{fs}$) to drive unconventional magnets, followed by an ultraviolet (UV) probe, conditions (i) and (ii) can be naturally satisfied. A pump pulse duration of order $10\text{T} \sim 1\,\text{ps}$ is sufficient to establish Floquet-induced spin splitting and Floquet pair correlations [Condition (ii)], while remaining much shorter than the lifetime of the prethermal regime, which can persist up to $\sim 10^4 T$ [242] [Condition (i)]. Moreover, solid-state systems are not perfectly isolated, and dissipation to the environment is unavoidable [245]. Energy absorbed from the driving field can be partially transferred to the substrate and other degrees of freedom, such as phonons, which further extends the lifetime of the prethermalized state and mitigates runaway heating [126, 242].

In summary, although thermalization and carrier redistribution pose intrinsic limitations to Floquet descriptions, their impact can be substantially reduced by choosing suitable driving pulse shapes and durations. Under these conditions, the system remains in a long-lived prethermal regime. While the precise form of the prethermal distribution remains an open problem [126], the Floquet formalism employed in the current manuscript captures the essential properties of the prethermal state and is supported by existing experimental results [130, 142, 230, 231, 242].

# 7    Conclusions

We have studied the effect of time-periodic light drives on $p$- and $d$-wave unconventional magnets with and without spin-singlet $s$-wave superconductivity. In particular, we have considered circularly and linearly polarized drives and uncovered a nontrivial light-matter interaction entirely tied to the way light interacts with unconventional magnetism. We then demonstrated that this nontrivial interaction generates Floquet spin density and Floquet spin-triplet Cooper pairs that do not exist in the static regime and can be used to identify the type of unconventional magnetism. Moreover, we found that the emergence of Floquet Cooper pairs, accompanied by a Floquet spin density, is intrinsic to the Floquet sidebands ,which provide an additional quantum number (the Floquet sideband index) that broadens the classification of superconducting symmetries into Floquet classes that coexist between different sidebands through the absorption or emission of photons. Particularly, we identified two spin-singlet Floquet pair classes that arise due to the time-periodic field and conventional superconductivity, while two types of spin-triplet Floquet Cooper pairs are generated by the combined effect of unconventional magnetism, conventional superconductivity, and time-periodic driving. As a result, the classification of the spin-triplet Floquet Cooper pairs highly depends on the distinct momentum parities inherited from the $p$- and $d$-wave unconventional magnets. We further showed that the distinct Floquet Cooper pairs can be controlled by exploiting the light polarization and the orientation of the unconventional magnetic order, offering a direct way to probe the symmetries of unconventional magnetism. Our findings demonstrate that unconventional magnets represent a versatile platform for realizing light-induced Cooper pairs by means of Floquet engineering. These results also raise natural questions, such as how robust are the light-induced superconducting pairs against realistic conditions where disorder [198, 220, 246–251] and dissipation [214, 252–255] are very likely effects that cannot be avoided; to answer these questions, a detailed investigation is needed.

# Acknowledgments

**Acknowledgments**   P.-H.F. thanks W. X. for helpful discussions and support.

**Author contributions**   P.-H. F. carried out the analytical and numerical calculations, made the figures, and prepared the manuscript. S.M. performed complementary calculations, contributed to data interpretation, and manuscript preparation. J.-F. L. supported the numerical calculations and provided comments on the manuscript. J. C. conceived the idea, supervised the project, contributed to the interpretation of results, and helped write the final version of the manuscript. All authors discussed the results, contributed to the writing of the manuscript, and approved its final version.

**Funding information**   S. M. and J. C. acknowledge financial support from the Göran Gustafsson Foundation (Grant No. 2216), the Carl Trygger's Foundation (Grant No. 22: 2093), and the Swedish Research Council (Vetenskapsrådet Grant No. 2021-04121). J.-F. Liu acknowledges financial support from the National Natural Science Foundation of China (Grant No. 12174077).

# A   Hamiltonian and Floquet components in higher-order-momentum unconventional magnet

To demonstrate a pedagogic example of the Floquet engineering unconventional magnet, $d$-wave and $p$-wave magnets are considered in the main text, see Eq. (4). Our results about the multiple spin degenerate nodes, Floquet spin density, and the driving induced Floquet Cooper pairs can be applied to all kinds of unconventional magnets, whose anisotropic spin split effect is captured by Eq. (2). For completeness, in this appendix, we present the Floquet components of all types of unconventional magnets subjected to CPL and LPL. By expanding Eq. (2), the explicit forms of the unconventional magnetic terms are given as:

$$
\begin{aligned}
J_0(\boldsymbol{k}) &= \alpha_s & s\text{-wave} \\
J_1(\boldsymbol{k}) &= \alpha_p(k_x \cos\theta_J + k_y \sin\theta_J) & p\text{-wave} \\
J_2(\boldsymbol{k}) &= \alpha_d \left[ (k_x^2 - k_y^2)\cos 2\theta_J + 2k_x k_y \sin 2\theta_J \right] & d\text{-wave} \\
J_3(\boldsymbol{k}) &= \alpha_f \left[ k_x(k_x^2 - 3k_y^2)\cos 3\theta_J + k_y(k_y^2 - 3k_x^2)\sin 3\theta_J \right] & f\text{-wave} \\
J_4(\boldsymbol{k}) &= \alpha_g \left[ (k_x^4 - 6k_x^2 k_y^2 + k_y^4)\cos 4\theta_J + 4k_x k_y(k_x^2 - k_y^2)\sin 4\theta_J \right] & g\text{-wave} \\
J_6(\boldsymbol{k}) &= \alpha_i \Big[ (k_x^6 - 15k_x^4 k_y^2 + 15k_x^2 k_y^4 - k_y^6)\cos 6\theta_J \\
&\quad + 2k_x k_y(3k_x^4 - 10k_x^2 k_y^2 + 3k_y^4)\sin 6\theta_J \Big] & i\text{-wave}
\end{aligned} \tag{A.1}
$$

Here, $J_0$ corresponds to the conventional Zeeman interaction, while $J_5$ is forbidden due to the incompatibility of five-fold rotational symmetry with crystalline symmetries in periodic lattices. The $d$-wave and $p$-wave cases are Eq. (4) in the main text.

Further classification within each wave type depends on the orientation of the magnetic lobes, parameterized by the angle $\theta_J$. For example, in the $f$-wave magnet, the term $J_3(\boldsymbol{k})$ corresponds to the $f_{x(x^2-3y^2)}$-wave configuration when $\theta_J = 0$, and to the $f_{y(y^2-3x^2)}$-wave configuration when $\theta_J = \pi/6$.

One can verify that along the momentum directions defined by $\theta_k = \theta_q + n\pi/q$ ($n \in \mathbb{Z}$), the unconventional magnetic effect is maximized, resulting in spin-splitting that depends solely on the radial momentum magnitude as $\alpha_q k^q$. In contrast, along directions $\theta_k = \theta_q + (2n+1)\pi/(2q)$,

the magnetic term vanishes identically, leading to spin-degenerate band structures. These nodal and anti-nodal structures are directly inherited from the symmetry of the underlying magnet.

By implementing the Floquet formalism in Eq. (32), the Floquet components of a CPL-driven unconventional magnet take the following forms:

$$H_0^{q,\sigma} = H_q + Bk_A^2, \tag{A.2}$$

$$H_{+n}^{q,\sigma} = Bk_A e^{i\eta\theta_k}\delta_{n,1} + \sigma\alpha_q e^{\eta i q\theta_J}\frac{q!}{2(q-n)!n!}\left(ke^{-i\eta\theta_k}\right)^{q-n}k_A^n, \tag{A.3}$$

where $H_0^{q,\sigma}$ includes the zero-photon contribution, incorporating a self-doping term $Bk_A^2$ that effectively shifts the chemical potential (which we gauge away in our analysis). The $n$-photon Floquet components $H_{\pm n}^{q,\sigma}$ describe light–matter interaction processes. In particular, all components with $|n| > 1$ are non-trivial and arise from the interplay between the external driving and the unconventional magnetism, encoding higher-order photon-assisted transitions unique to these systems. Similar results can be obtained in the superconducting states.

In the LPL-driven case, the Floquet components become complicated, which are

$$H_0^{q,\sigma} = H_q^\sigma + \frac{1}{2}Bk_A^2 + \sigma\alpha_q k_A^2 m_0^q, \tag{A.4}$$

$$H_{+1}^{q,\sigma} = Bk_A k\cos(\theta_k - \phi_A) + \sigma\alpha_q k_A m_1^q, \tag{A.5}$$

$$H_{+2}^{q,\sigma} = \frac{1}{4}Bk_A^2 + \sigma\alpha_q k_A^2 m_2^q, \tag{A.6}$$

$$H_{+3}^{q,\sigma} = \sigma\alpha_q k_A^3 \times \begin{cases} 0, & \text{others} \\ \frac{1}{8}\cos(3\theta_J - 3\phi_A), & f\text{-wave} \\ \frac{1}{2}k\cos(4\theta_J - 3\phi_A - \theta_k), & g\text{-wave} \\ \frac{5}{16}\left[8k^2\cos(6\theta_J - 3\phi_A - 3\theta_k) \right. \\ \left. \quad + 3k_A^2\cos(6\theta_J - 5\phi_A - \theta_k)\right], & i\text{-wave} \end{cases}, \tag{A.7}$$

$$H_{+4}^{q,\sigma} = \sigma\alpha_q k_A^4 \times \begin{cases} 0, & \text{others} \\ \frac{1}{16}\cos(4\theta_J - 4\phi_A), & g\text{-wave} \\ \frac{3}{32}\left[k_A^2\cos(6\theta_J - 6\phi_A) \right. \\ \left. \quad + 10k^2\cos(6\theta_J - 4\phi_A - 2\theta_k)\right], & i\text{-wave} \end{cases}, \tag{A.8}$$

$$H_{+5}^{q,\sigma} = \begin{cases} 0, & \text{others} \\ \frac{3}{16}\sigma\alpha_i k_A^5\cos(6\theta_J - 5\phi_A - \theta_k), & i\text{-wave} \end{cases}, \tag{A.9}$$

and

$$H_{+6}^{q,\sigma} = \begin{cases} 0, & \text{others} \\ \frac{1}{64}\sigma\alpha_i k_A^6\cos(6\theta_J - 6\phi_A), & i\text{-wave} \end{cases}, \tag{A.10}$$

with

$$m_0^q = \begin{cases} 0, & p\text{-wave} \\ \frac{1}{2}\cos(2\theta_J - 2\phi_A), & d\text{-wave} \\ \frac{3}{2}k\cos(3\theta_J - 2\phi_A - \theta_k), & f\text{-wave} \\ \frac{3}{8}\left[k_A^2\cos(4\theta_J - 4\phi_A) + 8k^2\cos(4\theta_J - 2\phi_A - 2\theta_k)\right], & g\text{-wave} \\ \frac{5}{16}\left[k_A^4\cos(6\theta_J - 6\phi_A) + 24k^4\cos(6\theta_J - 2\phi_A - 4\theta_k) \right. \\ \left. \quad + 18k_A^2 k^2\cos(6\theta_J - 4\phi_A - 2\theta_k)\right], & i\text{-wave} \end{cases}, \tag{A.11}$$

$$
m_1^q = \begin{cases}
\frac{1}{2}\cos(\theta_J - \phi_A), & p\text{-wave} \\
k\cos(2\theta_J - \phi_A - \theta_k), & d\text{-wave} \\
\frac{3}{8}\left[k_A^2 \cos(3\theta_J - 3\phi_A) + 4k^2 \cos(3\theta_J - \phi_A - 2\theta_k)\right], & f\text{-wave} \\
\frac{1}{2}k\left[4k^2 \cos(4\theta_J - \phi_A - 3\theta_k) + 3k_A^2 \cos(4\theta_J - 3\phi_A - \theta_k)\right], & g\text{-wave} \\
\frac{3}{8}k\left[8k^4 \cos(6\theta_J - \phi_A - 5\theta_k) + 20k_A^2 k^2 \cos(6\theta_J - 3\phi_A - 3\theta_k)\right. \\
\qquad \left. + 5k_A^2 \cos(6\theta_J - 5\phi_A - \theta_k)\right], & i\text{-wave}
\end{cases},
$$

and

$$
m_2^q = \begin{cases}
0, & p\text{-wave} \\
\frac{1}{4}\cos(2\theta_J - 2\phi_A), & d\text{-wave} \\
\frac{3}{4}k\cos(3\theta_J - 2\phi_A - \theta_k), & f\text{-wave} \\
\frac{1}{4}\left[k_A^2 \cos(4\theta_J - 4\phi_A) + 6k^2 \cos(4\theta_J - 2\phi_A - 2\theta_k)\right], & g\text{-wave} \\
\frac{15}{64}\left[k_A^2 \cos(6\theta_J - 6\phi_A) + 16k^2 \cos(6\theta_J - 2\phi_A - 4\theta_k)\right. \\
\qquad \left. + k_A^2 (6\theta_J - 4\phi_A - 2\theta_k)\right], & i\text{-wave}
\end{cases}.
\qquad (A.12)
$$

Among all Floquet components in the LPL-driven case, particular attention is drawn to the zero-photon sector in Eq. (A.4), which contains an effective magnetic contribution encoded in $m_0^q$ [see Eq. (A.11)]. This term reveals that light-induced corrections to the static Hamiltonian can emulate effective unconventional magnetic fields of different symmetry classes, depending on the underlying magnet and the polarization direction $\phi_A$. For example, in an LPL-driven $i$-wave magnet, the zero-photon contribution includes effective magnetic terms such as a $g$-wave-like component $k^4 \cos(6\theta_J - 2\phi_A - 4\theta_k)$, an $f$-wave-like component $k^2 \cos(6\theta_J - 4\phi_A - 2\theta_k)$, and an $s$-wave-like term $k_A^4 \cos(6\theta_J - 6\phi_A)$. The relative strengths of these components can be selectively tuned by adjusting the light intensity $k_A$ and the linear polarization angle $\phi_A$, thereby enabling the emulation of lower-order unconventional magnets from higher-order ones. This mechanism offers a route for engineering effective magnetism with tailored spatial symmetries via Floquet driving, even in the absence of those magnetic orders in the static system [75].

# B  Effect of the strength of $d$-wave magnet $\alpha_d$

The results obtained in the driven normal state for the spin density in Sec. 4 and in the superconducting state for the Cooper pair amplitude in Sec. 5 are based on a $d$-wave magnet with $\alpha_d < B$, which is referred to as a weak $d$-wave magnet [37, 75]. Such a system hosts two upward-opening parabolas with spin- and direction-dependent curvature in the dispersion, resulting in two closed spin-dependent elliptical Fermi surfaces with orthogonal principal axes [37]. Our numerical results confirm that all results discussed above remain valid for the *strong $d$*-wave magnet with $\alpha_d \geq B$, where the dispersion features a saddle point at $(k_x, k_y) = (0,0)$ and spin-dependent bands that open in opposite directions.

In the driven strong $d$-wave magnet with arbitrary $\theta_J$, our numerical results confirm that the main findings persist, including the presence of multiple spin-degenerate nodes in the Floquet spin density, the structure of the Floquet BdG spectrum, the symmetry classification of the Cooper pair amplitude, the contributions from various photon processes, and the effects of linearly polarized light. This indicates that our results are universal for $d$-wave magnets of arbitrary strength.

In summary, both weak and strong $d$-wave magnets exhibit the same rich Floquet-engineered phenomena, underscoring the generality and robustness of the symmetry-based classification and light-induced effects presented in this work.

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
