# Peer review of "Light-induced Floquet spin-triplet Cooper pairs in unconventional magnets"

_SciPost Physics_

## Round 1 · Referee Report · Anonymous (Referee 1) · 2025-11-4

Strengths

1.the article is very well written
2.easy to follw
3.pleasant to read
4.the idea behind it is intriguing
5.the results are rich and well presented
6.the formalism used is described in a transparent and informative way.

Weaknesses

1.no discussion on the observability of the effects is given, in terms of magnitude of the relevant parameters
2.the limitations of the formalism are not stated (what about heating, Floquet band occupation,..)

Report

In their article, the authors address the properties of irradiated non-conventional magnets, both with and without superconductivity. Their analysis is deep and exhaustive, the results are convincing and, to the best of my knowledge, novel. The authors still need to better address the experimental significance of their results. Once this is done, I recommend the article for publication.

Requested changes

Add a discussion about the observability of the effects.

Recommendation

Publish (easily meets expectations and criteria for this Journal; among top 50%)

---

## Round 1 · Referee Report · Anonymous (Referee 2) · 2025-11-26

Report

In this manuscript, the authors address how time-periodic light drives (circularly and linearly polarized light) induce Floquet spin-triplet densities and Floquet spin-triplet Cooper pairs in unconventional magnets with $p$- and $d$-wave parity, especially $p_{x}$ and $d_{x^2-y^2}$. Using Floquet theory and BdG formalism, the authors identify photon-induced processes that generate new pairing symmetries absent in equilibrium. The topic is timely and relevant to Floquet-engineered quantum systems and unconventional magnetism.

In the appendix, the authors provide a section entitled “Hamiltonian and Floquet components in higher-order-momentum unconventional magnet.” I really appreciate this section.

The manuscript is well written and technically solid, but several points require clarification or further justification before publication.

Major and minor comments:

  1. Could the author explain how to write the effective model involving two arbitrary Floquet sidebands, i.e., Eq.(39)?

  2. How many Floquet sidebands were kept in the calculations, and how did the authors ensure convergence of the spin density and pairing amplitudes?

  3. Could the authors comment on whether the Floquet triplet states would survive in a realistic Floquet heating effect?

  4. Is there any simple physical explanation for why odd-photon processes produce purely light-induced pairing channels, while even-photon processes can mix with static correlations? Is it stable against disorder?

  5. Resolution of a few figures (Fig. 3 and 4, for example) should be increased for clarity.

  6. There are a few recent studies on Floquet in altermagnetic systems [eg, Phys. Rev. B 112, L201408 (2025)]. The author should cite those articles accordingly.

  7. Is it possible to relate Floquet spin-triplet density to any experimental observable?

Typos to be fixed:

  1. In page number 7, below Eq.(4), they have written "while a $p_x$-wave magnet for $\theta_J = \frac{\pi}{2}$." It should be "while a $p_y$-wave magnet for $\theta_J = \frac{\pi}{2}$."

  2. In Eq.(40), the authors should keep space in between 2 and $\delta n$ and no space between $\delta$ and $n$. The correct form seems to be $\delta$ times $n$, rather than $\delta n$.

  3. In Fig. 4, panels (b) and (c) should be interchanged to be consistent with the figure caption and explanation in the main text.

  4. A few spelling typos are also there.

Requested changes

Modify the figures and add possible explanations according to the report.

Recommendation

Ask for minor revision

---

## Editorial Decision

awaiting_resubmission